# Semidefinite Relaxations of the Gromov-Wasserstein Distance

Junyu Chen [*]    Binh T. Nguyen [*]    Shang Hui Koh    Yong Sheng Soh

Department of Mathematics
National University of Singapore
chenjunyu@u.nus.edu,binhnt@nus.edu.sg,matsys@nus.edu.sg

## Abstract

The Gromov-Wasserstein (GW) distance is an extension of the optimal transport problem that allows one to match objects between incomparable spaces. At its core, the GW distance is specified as the solution of a non-convex quadratic program and is not known to be tractable to solve. In particular, existing solvers for the GW distance are only able to find locally optimal solutions. In this work, we propose a semi-definite programming (SDP) relaxation of the GW distance. The relaxation can be viewed as the Lagrangian dual of the GW distance augmented with constraints that relate to the linear and quadratic terms of transportation plans. In particular, our relaxation provides a tractable (polynomial-time) algorithm to compute globally optimal transportation plans (in some instances) together with an accompanying proof of global optimality. Our numerical experiments suggest that the proposed relaxation is strong in that it frequently computes the globally optimal solution. Our Python implementation is available at https://github.com/tbng/gwsdp.

## 1   Introduction

The optimal transport (OT) problem concerns the task of finding a transportation plan between two probability distributions to minimize some costs. The problem has applications in a wide range of scientific and engineering applications. For instance, in the context of machine learning, the OT problem forms the backbone of recent breakthroughs in generative modeling (Arjovsky et al., 2017; Liu et al., 2022; Lipman et al., 2022), natural language processing (Kusner et al., 2015), domain adaptation (Courty et al., 2017), and single-cell alignment (Schiebinger et al., 2019; Bunne et al., 2023, 2024).

Let $\alpha \in \Sigma_m$ and $\beta \in \Sigma_m$ be discrete probability distributions over a metric space – here $\Sigma_m := \{\alpha \in \mathbb{R}_+^m, \sum_{i=1}^m \alpha_i = 1\}$ denotes the probability simplex. Let $C \in \mathbb{R}^{m \times n}$ be the matrix such that $C_{i,j}$ models the transportation cost between points $x_i \sim \alpha$ and $y_j \sim \beta$. The (Kantorovich) formulation of the discrete OT problem (Kantorovich, 1942; Villani et al., 2009; Santambrogio, 2015; Peyré et al., 2019) is defined as the solution of the following convex optimization instance

$$\pi_{\mathcal{W}} \overset{\text{def.}}{=} \underset{\pi \in \Pi(\alpha,\beta)}{\operatorname{argmin}} \langle C, \pi \rangle. \tag{1}$$

Here, $\Pi(\alpha, \beta) = \{\pi \in \mathbb{R}_+^{m \times n} : \pi \mathbb{1}_n = \alpha, \pi^\top \mathbb{1}_m = \beta\}$ denotes the set of couplings between probability distributions $\alpha, \beta \in \Sigma_m$, while $\mathbb{1}_m \in \mathbb{R}^m$ denotes the vector of ones. The OT problem (1) is an instance of a linear program (LP), and hence admits a global minimizer.

---

[*]Equal contribution.

One limitation of the classical OT formulation in (1) is that the definition of the cost matrix $C$ requires the probability distributions $\alpha$ and $\beta$ to reside in the same metric space. This is problematic in application domains where we wish to compare probability distributions in different spaces, in which case there is no meaningful way to describe the cost of moving from one location to another. Such settings arise frequently in shape comparison and graph matching applications, for example.

To address such scenarios, the work of Mémoli (2011) formulates an extension of the OT problem known as the Gromov-Wasserstein distance (GW) whereby one can define an analogous OT problem given knowledge of the cost matrices for the respective spaces where $\alpha$ and $\beta$ reside in. More concretely, let the tuple $(C, \alpha) \in \mathbb{R}^{m \times m} \times \Sigma_m$ denote a discrete metric-measure space. Given a smooth, differentiable function $\ell : \mathbb{R} \times \mathbb{R} \to \mathbb{R}$, the Gromov-Wasserstein distance between two discrete metric-measure spaces $(C, \alpha)$ and $(D, \beta)$ is defined by

$$\text{GW}(C, D, \alpha, \beta) \overset{\text{def.}}{=} \min_{\pi \in \Pi(\alpha, \beta)} \ell(C_{i,k}, D_{j,l}) \pi_{i,j} \pi_{k,l} = \min_{\pi \in \Pi(\alpha, \beta)} \langle \mathbf{L}(C, D) \otimes \pi, \pi \rangle. \quad \text{(GW)}$$

Here, the transportation cost is specified by the four-way tensor that measures the discrepancy between the metrics $C$ and $D$

$$\mathbf{L}(C, D)_{i,j,k,l} \overset{\text{def.}}{=} \ell(C_{i,k}, D_{j,l}). \quad (2)$$

The squared loss error, for instance, is a common choice. Following Peyré et al. (2016), we define the tensor-matrix multiplication by

$$[\mathbf{L} \otimes \pi]_{i,j} \overset{\text{def.}}{=} \sum_{k,l} \mathbf{L}_{i,j,k,l} \pi_{k,l}.$$

The GW distance has been applied widely to machine learning tasks, most notably on graph learning (Vayer et al., 2019a; Xu et al., 2019; Vincent-Cuaz et al., 2021, 2022). It is an instance of a quadratic program (QP) – these are optimization instances in which we minimize a quadratic objective subject to some linear inequalities. To see this, one can re-write the objective in (GW) in terms of vectorized matrices

$$\min_{\pi} \ \langle \text{vec}(\pi), L \, \text{vec}(\pi) \rangle \ \text{ s.t. } \ \pi \in \Pi(\alpha, \beta). \quad \text{(GW+)}$$

Here, $(L_{ij,kl})_{ij,kl} \in \mathbb{R}^{mn \times mn}$ denotes the flattened 2-dimension tensor of $\mathbf{L}$, while the vectorization of a matrix $\pi \in \mathbb{R}^{m \times n}$ is given by

$$\text{vec}(\pi) \overset{\text{def.}}{=} [\pi_{11}, \pi_{21}, \ldots, \pi_{m1}, \ldots, \pi_{mn}]^\top \in \mathbb{R}^{mn}.$$

The constraint $\pi \in \Pi(\alpha, \beta)$ is convex, and in fact linear. On the other hand, the matrix $L$ need not be positive semidefinite, and as such, the QP instance in (GW+) is non-convex in general. In fact, in the cases where the matrix $L$ arises as the difference of cost matrices (2), $L$ is *never* positive semidefinite as these are zero on the diagonal while the off-diagonal entries are non-negative.

## 2 Main Contributions

The main contribution of this work is to propose a strong semidefinite programming (SDP)-based relaxation for the Gromov-Wasserstein distance that leads to globally optimal solutions in many instances. Concretely, let $(\pi_{sdp}, P_{sdp})$ denote an optimal solution to the following

$$
\begin{aligned}
\text{GW-SDP}(C, D, \alpha, \beta) \quad &\overset{\text{def.}}{=} \min_{\substack{\pi \in \mathbb{R}^{m \times n}, \\ P \in \mathbb{R}^{mn \times mn}}} \quad \langle L, P \rangle \\
&\text{s.t.} \quad \begin{pmatrix} P & \text{vec}(\pi) \\ \text{vec}(\pi)^\top & 1 \end{pmatrix} \succeq 0 \\
&\quad \pi \in \Pi(\alpha, \beta) \\
&\quad P \text{vec}(e_i \mathbb{1}_n^\top) = \alpha_i \text{vec}(\pi), i \in [m] \\
&\quad P \text{vec}(\mathbb{1}_m e_j^\top) = \beta_j \text{vec}(\pi), j \in [n] \\
&\quad P \geq 0
\end{aligned}
\quad \text{(GW-SDP)}
$$

Here, $e_i$ denotes the standard basis vector whose $i$-th entry is $1$. This relaxation can be viewed as the Lagrangian dual of the GW problem augmented with constraints that relate the linear and quadratic

terms of transportation plans (we discuss these aspects in greater detail in Appendix B). A simpler way to express the condition $P\text{vec}(e_i \mathbb{1}_n^\top) = \alpha_i \text{vec}(\pi)$ is to note that

$$P\text{vec}(e_i \mathbb{1}_n^\top) = \alpha_i \text{vec}(\pi) \quad \Leftrightarrow \quad \sum_j P_{(i,j),(k,l)} = \alpha_i \pi_{k,l},$$

$$P\text{vec}(\mathbb{1}_m e_j^\top) = \beta_j \text{vec}(\pi) \quad \Leftrightarrow \quad \sum_i P_{(i,j),(k,l)} = \beta_j \pi_{k,l}.$$

We begin by noting a basic observation: Let $\pi \in \Pi(\alpha, \beta)$ be a transportation plan. Then the tuple $(\pi, P) = (\pi, \text{vec}(\pi)\text{vec}(\pi)^\top)$ is a feasible solution to (GW-SDP) since $\text{vec}(\pi)\text{vec}(\pi)^\top \text{vec}(e_i \mathbb{1}_m^\top) = \text{vec}(\pi)\langle \pi, e_i \mathbb{1}_m^\top \rangle = \text{vec}(\pi)\langle \pi \mathbb{1}_n, e_i \rangle = \alpha_i \text{vec}(\pi)$. The inequalities for $\beta$ follow analogously. This implies that the optimal value of (GW-SDP) is a lower bound to the GW problem (GW): Let $\pi^\star$ denote an optimal solution to the GW problem (note: this is (GW), which is equivalent to (GW+)). By recalling that the tuple $(\text{vec}(\pi^\star), \text{vec}(\pi^\star)\text{vec}(\pi^\star)^\top)$ is a feasible solution to (GW-SDP), one has

$$\langle P_{sdp}, L \rangle \leq \langle \text{vec}(\pi^\star), L \,\text{vec}(\pi^\star) \rangle = \langle \pi^\star, \mathbf{L} \otimes \pi^\star \rangle. \tag{3}$$

The inequality (3) provides us with a principled way of *certifying* global optimality of a given transportation plan. Let $\pi \in \Pi(\alpha, \beta)$ be an arbitrary transportation plan. A natural approach to quantify the quality of $\pi$ is to compare its objective value with the optimal choice:

$$\text{Apx. Ratio}(\pi) := \frac{\langle \pi, \mathbf{L} \otimes \pi \rangle}{\langle \pi^\star, \mathbf{L} \otimes \pi^\star \rangle}.$$

This ratio is at least one and is equal to one if $\pi$ is also globally optimal. A consequence of (3) is the following upper bound

$$\text{Apx. Ratio}(\pi_{sdp}) \leq \frac{\langle \pi_{sdp}, \mathbf{L} \otimes \pi_{sdp} \rangle}{\langle P_{sdp}, L \rangle}. \tag{4}$$

Note that all the quantities in the RHS can be computed efficiently as the solution of a SDP. Suppose we are able to do so, and in the process evaluate the RHS to be equal to one. *Then, we have a proof that $\pi_{sdp}$ is the global optimal solution to the GW problem.* In a recent work that appeared during the reviewing process of our work, the GW problem is shown to be intractable in general (Kravtsova, 2024). What our discussion shows is that it is possible, in some instances, to obtain the globally optimal solution via a polynomial-time algorithm by solving (GW-SDP), and with a guarantee that the obtained solution is indeed globally optimal. In fact, the instances for which one can obtain an upper bound equal to one using (GW-SDP) is not as far-fetched as one might think: our numerical experiments in Section 4 show that this happens quite often, and especially so whenever $m = n$. **No restrictions on cost tensor.** One of the strengths of our proposed SDP relaxation is that it is valid for *all* cost tensors $\mathbf{L}$. This stands in contrast with other methods like the entropic GW (Peyré et al., 2016), which is only applicable to the cost that can be decomposed to a specific form such as the $\ell_2$ or discrete KL loss.

## 3    SDP Relaxations of QPs

We motivate the relaxation in (GW-SDP). The starting point is to recognize that the GW problem is an instance of a QPs – these are optimization instances of the form

$$\min_{x \in \mathbb{R}^n} \quad x^\top A x + 2b^\top x + c \qquad \text{s.t.} \qquad Bx \leq d. \tag{5}$$

QPs are an important class of optimization problems. If the matrix $A$ is PSD, then the objective is convex, and the QP instance can be solved tractably using standard software (Nocedal and Wright, 2006). The problem becomes difficult if $A$ contains negative eigenvalues. The general class of QPs is NP-hard; for instance, it contains the problem of finding the maximum clique of a graph (Motzkin and Straus, 1965). In fact, the presence of a *single* negative eigenvalue in $A$ is sufficient to make the class of QPs NP-hard (Pardalos and Vavasis, 1991). The typical approach to solving a quadratic program exactly is via a branch-and-bound type of algorithm. Other approaches include relating QPs to the class of co-positive programming, mixed integer linear programming, and deploying SDP relaxations (Bomze and de Klerk, 2002) – typically, these methods are used as a sub-routine within a branch-and-bound procedure.

**Standard SDP Relaxation.** The first step of SDP relaxation is to express the quadratic terms with a PSD matrix whose rank is one. Concretely, the QP instance (5) is equivalent to the following:

$$\min_{x \in \mathbb{R}^n, X \in \mathbb{R}^{n \times n}} \quad \text{tr}(AX) + 2b^\top x + c$$
$$\text{s.t.} \quad Bx \leq d$$
$$\begin{pmatrix} X & x \\ x^\top & 1 \end{pmatrix} \succeq 0, \ \ \text{rank}\begin{pmatrix} X & x \\ x^\top & 1 \end{pmatrix} = 1$$

This optimization instance is not convex because of the rank-one constraint. The second step is to simply omit the rank constraint, which yields a semidefinite program and therefore is convex. This is the standard SDP relaxation for QPs (the technique applies more generally to quadratically constrained quadratic programs – QCQPs).

By applying the same sequence of steps to (GW+), the standard SDP relaxation one arrives at is the following:

$$\min_{\pi \in \mathbb{R}^{m \times n}, P \in \mathbb{R}^{mn \times mn}} \quad \langle L, P \rangle$$
$$\text{s.t.} \quad \begin{pmatrix} P & \text{vec}(\pi) \\ \text{vec}(\pi)^\top & 1 \end{pmatrix} \succeq 0 \quad (6)$$
$$\pi \in \Pi(\alpha, \beta)$$

Problem (6) is a tractable convex semidefinite programming, which can be efficiently solved in polynomial time. If the solution to (6) (and the subsequent SDP relaxations we introduce) has a rank equal to one, we would have solved the original GW problem (GW). Unfortunately, the feasible region of $P$ in (6) is not compact, and the optimal value to (6) is unbounded below in general.

**Proposition 3.1.** *The optimization instance* (6) *is unbounded below.*

**Tightening the Relaxation.** As such, it is necessary to augment (6) with additional constraints to further strengthen the relaxation. Recall that the relaxation (6) is exact if $P = \text{vec}(\pi)\text{vec}(\pi)^\top$. Therefore, a simple way to improve the relaxation is to add any linear constraints that is satisfied by solutions of the form $(\pi, P) = (\pi, \text{vec}(\pi)\text{vec}(\pi)^\top)$.

First, $\pi \geq 0$ for all $\pi \in \Pi(\alpha, \beta)$, and hence $\text{vec}(\pi)\text{vec}(\pi)^\top \geq 0$. This means we may freely impose

$$P \geq 0. \tag{Nng.}$$

Second, note that $\sum_i \pi_{ij}\pi_{kl} = \pi_{kl}(\sum_i \pi_{ij}) = \pi_{kl}\beta_j$. Subsequently, we may impose $\sum_i P_{(i,j),(k,l)} = \beta_j \pi_{kl}$. This leads to the following set of equalities:

$$P\text{vec}(e_i \mathbb{1}_n^\top) = \alpha_i \text{vec}(\pi), i \in [m], \qquad P\text{vec}(\mathbb{1}_m e_j^\top) = \beta_j \text{vec}(\pi), j \in [n]. \tag{Mar.}$$

The proposed SDP relaxation (GW-SDP) is precisely (6) with the additional constraints (Nng.) and (Mar.). In addition, the set of matrices $P$ satisfying (Mar.) have trace at most one. Hence the feasible region is a subset of PSD matrices with trace at most one, which is compact.

**Relation to the QAP.** We point out that the constraints (Nng.) and (Mar.) have been previously proposed for a different but closely related problem known as Quadratic Assignment Problem (QAP) (Dym et al., 2017; Kezurer et al., 2015; Zhao et al., 1998). Mathematically, the QAP problem can be viewed as equivalent to (GW+) but with the additional restriction that $m = n$ and that $\pi$ is a *permutation* matrix. The work in Zhao et al. (1998) proposes a SDP relaxation that is effectively equivalent to (GW-SDP) but with additional linear equalities implied by orthogonality $\pi\pi^\top = \pi^\top\pi = I$. The works in Dym et al. (2017); Kezurer et al. (2015) subsequently build on the ideas in Zhao et al. (1998) and propose more scalable alternatives while providing tight relaxations.

The key difference between the QAP and the GW problem we investigate is that $\pi$ is not necessarily a permutation matrix in the GW problem, and necessarily so if $m \neq n$. As such, the relaxation in Zhao et al. (1998) is invalid. Our contribution is to recognize that, by omitting the constraints corresponding to orthogonality, one obtains an SDP relaxation that now becomes valid for the GW problem, which leads to good practical performance.

**No need for rounding.** One important property of the relaxation in (GW-SDP) is that the output will always be a feasible transportation plan in $\Pi(\alpha, \beta)$. This means that no additional rounding

is necessary. This is vastly different from combinatorial optimization problems including the QAP where the optimal solution to the relaxation is not guaranteed to be a feasible solution, and additional rounding steps may be necessary.

# 4  Numerical Experiments with Off-the-shelf Convex Solvers

In this section, we implement our proposed SDP relaxation using an off-the-shelf solver. We compare our method with the Conditional Gradient (CG-GW) solver for finding local solutions (Vayer et al., 2019a), and the Sinkhorn projections solver for computing solutions to the entropic GW (eGW) problem by Peyré et al. (2016). Both of the latter are implemented in the Python Optimal Transport library (PythonOT, Flamary et al. 2021). The goal is to show that our proposed SDP relaxation frequently computes the global optimal transportation plan whereas existing methods frequently do not.

In what follows, we will use the 2-Gromov-Wasserstein distance, *i.e.* the cost function is squared Euclidean norm. We solve the GW-SDP instance implemented in CVXPY (Diamond and Boyd, 2016) using the SCS and MOSEK solvers (ApS, 2022; O'Donoghue et al., 2016).

## 4.1  Matching Gaussian Distributions

In this example, we estimate the GW distance between two Gaussian point clouds, one in $\mathbb{R}^2$, and the other in $\mathbb{R}^3$. A visualization of this dataset can be found in Figure 1a. The classical optimal transport formulation such as the likes of Wasserstein-2 distance does not apply because the two point clouds belong to different spaces.

As seen in a qualitative demonstration of Figure 1b, our algorithm returns optimal transport plans that are as sparse as the transportation plans obtained via the Conditional Gradient descent solver of Python OT for GW distance (CG-GW). We also vary the number of sample points and calculate the value of the objective function $\langle \pi, \mathbf{L} \otimes \pi \rangle$. As shown in Figure 2a, the transport plans obtained by (GW-SDP) consistently returns smaller objective value (orange line) than those obtained via the GW-CG counterpart from PythonOT (blue line) and its entropic regularization (green line). This shows that the transport plans computed by PythonOT, for instance, are in fact frequently sub-optimal.

In Figure 2b, we plot the estimated approximation ratio across different numbers of sample points. We notice that in this scenario of Gaussian matching, the estimated approximation ratio is close to 1.0 in most instances – this tells us that the (GW-SDP) frequently computes globally optimal transportation plans. In contrast, local methods such as PythonOT often do not. Note that we also observed the sparsity of the SDP-GW transport plans in this varying scenario.

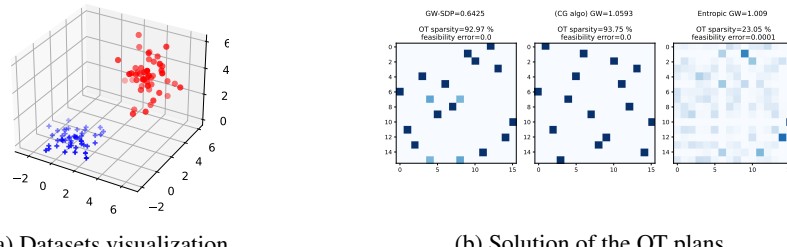

(a) Datasets visualization            (b) Solution of the OT plans

Figure 1: **Left:** source distribution (2D, blue dots) and target distribution (3D, red dots). For ease of visualization, we lift the source $\mathbb{R}^2$ mm-spaces into target $\mathbb{R}^3$ by padding the third coordinate to zero. **Right:** OT solutions of GW-SDP (our algorithm), CG-GW (conditional gradient descent, default solver of PythonOT) and entropic OT solver. The OT plans from GW-SDP is almost sparse in the same manner to CG-GW, while the eGW is not.

**Scenario where m $\neq$ n.** The bulk of our experiments focus on the setting where $m = n$. We performed an experiment where $m \neq n$: the number of samples in one distribution is fixed ($n = 8$) and we vary the number of samples $m$ in the other distribution. From our results in Figure 3, we notice that the relaxation is exact whenever $m$ is a multiple of $n$. On the other hand, when $m$ is not a multiple of $n$, we still observe exactness, but much less frequently.

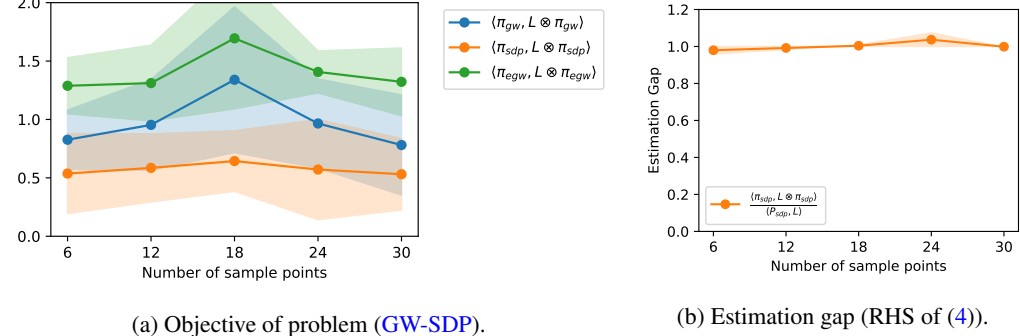

(a) Objective of problem (GW-SDP).

(b) Estimation gap (RHS of (4)).

Figure 2: Value of the objective (left) and approximation ratio (right) with a varying number of sample points, calculated on 10 runs of the Gaussian matching experiment.

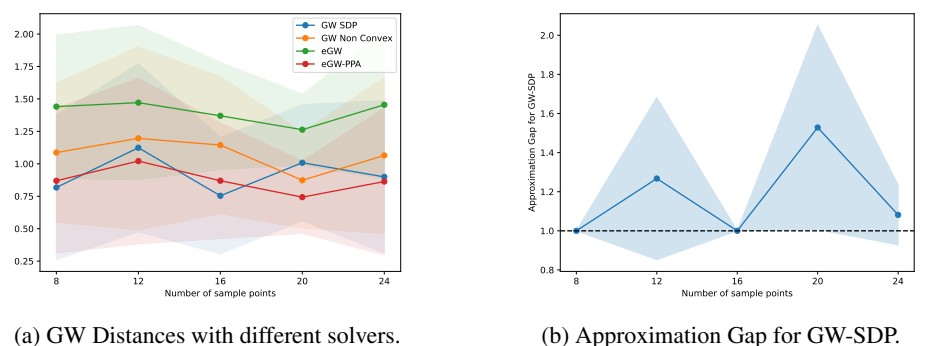

(a) GW Distances with different solvers.

(b) Approximation Gap for GW-SDP.

Figure 3: Gaussian Matching experiments with the sample points $m$ from source distribution varying while the sample points from target distribution keeping fixed $n = 8$. Average of 20 runs.

**Runtime Comparisons**  Table 1 presents the run-time of the GW-SDP problem in Experiment 1, running on a PC with 8 cores CPU and 32GB of RAM. In these experiments, the cost matrix $C$ is pre-computed (i.e. assumed given). As such, the run-time is independent of the data dimension. The GW-SDP has a matrix of dimension $mn \times mn$, which is slower than most local and entropic solvers. However, the solvers we implement (SCS and MOSEK) are off-the-shelf and are general SDP solvers that do not exploit special structures in the problem and do not provide options to use initialization of the transport plans. (SCS is a first-order method, but we are otherwise unaware of its complexity). We want to emphasize that in most settings where SDPs are applied, one will always try to develop specialized solvers that exploit the structure of the problem. In our setup, the optimal solution has low rank, and is rank-one if the relaxation is tight. There are numerous well-established methods for exploiting such structure. This is the subject of ongoing work.

Table 1: Average run-time in seconds for experiment in Figure 1a (matching Gaussians with varying number of samples $n$).

| n | **GW-SDP** | **GW-CG** | **eGW** ($\varepsilon = 0.1$) |
|---|---|---|---|
| 6 | 0.2437 (0.0265) | 0.0005 (0.000041) | 0.226 (0.1145) |
| 12 | 11.615 (2.4088) | 0.0006 (0.00003) | 0.2596 (0.0726) |
| 20 | 216.3645 (14.1123) | 0.0014 (0.000017) | 0.4923 (0.1500) |

**Comparisons of GW-SDP solver and GW-CG solver when number of sample points n increase.** We increase the number of samples for GW-CG (non-convex GW solver using conditional gradient descent or Frank-Wolfe algorithm) vs our GW-SDP solver for a fixed number of samples. From Table 2, we noticed that the objective value for GW-CG decreases as we increase the number of samples. For 100000 sample points, the GW-CG algorithm is more expensive and has a poorer

objective value than our method with 10 sample points. This suggests that our method can give good approximations of the GW distance with fewer sample points than existing methods.

Table 2: Comparisons of GW-SDP solver and GW-CG solver with a varying number of sample points $n$.

| n | GW-SDP | GW-SDP Runtime (s) | GW-CG | GW-CG runtime (s) |
|---|---|---|---|---|
| 10 | 0.4577 | 6.3753 | 1.135940 | 0.000389 |
| 100 | | | 0.629425 | 0.007571 |
| 1000 | | | 0.540984 | 2.520011 |
| 10000 | | | 0.496796 | 138.358954 |

## 4.2 Graph Community Matching

The objective of this task is to find matching between two random graphs that are drawn from the stochastic block model (SBM) (Abbe, 2017; Holland et al., 1983) with fixed inter/intra-clusters probability (the probability that nodes inside and outside a cluster are connected, respectively). The source is a three-cluster SBM whose intra-cluster probability is $p = \{1.0, 0.95, 0.9\}$, and the target is a two-cluster SBM whose intra-cluster probability is $p = \{1.0, 0.9\}$. The inter-clusters probability is all set to 0.1. The distance matrices on each graph are created first by simulating the node features drawn from Gaussian distributions with uniform weights. Subsequently, we compute the $\ell_2$ norm between nodes and shrink the value of disconnected nodes to zero to form the distance matrices.

We compare the transportation plans obtained using our methods with the baseline comparisons GW-CG and eGW in Figure 4a. We note that the (GW-SDP) model typically returns a transport plan with a smaller total transportation cost (i.e., a smaller objective value) $\langle \pi_{sdp}, \mathbf{L} \otimes \pi_{sdp} \rangle$. This trend is consistent with our observations in the previous experiments. Nevertheless, we see a degree of similarity between the transportation plans provided as output by all three methods. In addition, the transportation plans computed by our method and GW-CG are both reasonably sparse. This fact is observed in multiple runs of different seeds and graph sizes.

## 4.3 Extension of GW-SDP to Structured Data

In this example, we consider an extension of the (GW-SDP) to structured data, more specifically graphs with node features similar to the Fused-GW distance in (Vayer et al., 2019a). The discrete metric-measure space is now described by the tuple $(F, C, \alpha) \in \mathbb{R}^{m \times d} \times \mathbb{R}^{m \times m} \times \Sigma_m$, where $F \overset{\text{def.}}{=} (f_i)_i \in \mathbb{R}^d$ encodes the feature information of the sample point. The Fused GW-SDP (FGW-SDP) formulation is given by

$$\text{FGW-SDP}(M_{FG}, C, D, \alpha, \beta, \xi) \overset{\text{def.}}{=} \min_{\substack{\pi \in \mathbb{R}^{m \times n}, \\ P \in \mathbb{R}^{mn \times mn}}} \quad (1 - \xi)\langle M_{\alpha, \beta}, \pi \rangle + \xi \langle \mathbf{L}(C, D), P \rangle$$

$$\text{s.t.} \quad \begin{pmatrix} P & \text{vec}(\pi) \\ \text{vec}(\pi)^\top & 1 \end{pmatrix} \succeq 0 \qquad \text{(FGW-SDP)}$$

$$\pi \in \Pi(\alpha, \beta)$$

$$P\text{vec}(e_i \mathbb{1}_n^\top) = \alpha_i \text{vec}(\pi), i \in [m]$$

$$P\text{vec}(\mathbb{1}_m e_j^\top) = \beta_j \text{vec}(\pi), j \in [n]$$

$$P \geq 0,$$

with $M_{FG} = d(f_j, g_j)_{i,j}$ encodes the distance between node features, and $\xi \in [0, 1]$ the interpolation parameter. Figure 4b shows the result of matching two SBM graphs with the same setting as in Section 4.2, with the exception that now we input the feature to calculate $M_{FG}$ by $\ell_2$ norm, and the structured matrices are the shortest path matrices obtained from the adjacency matrices of the graphs. We set $\xi = 0.8$ for this example. The figure shows that the output OT plans and values of (FGW-SDP) and FGW-CG (using PythonOT) are identical, while entropic Fused-GW returned a higher value and a denser transport plan. This indicates that the SDP relaxation of Fused-GW can be useful in graph matching applications, akin to Fused-GW.

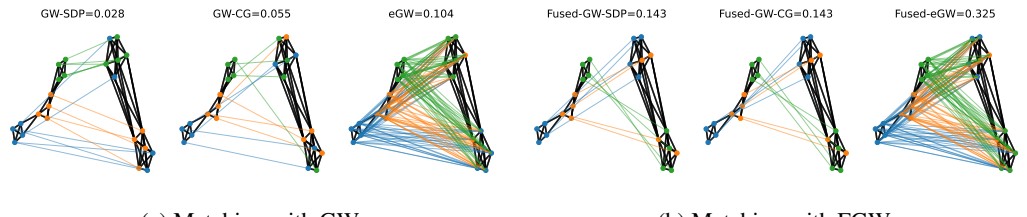

| (a) Matching with GW. | (b) Matching with FGW. |

Figure 4: Value of the objective on the synthetic graph matching task, from the three-block SBM (left) to the two-block SBM (right). **Upper:** calculated using GW. **Lower:** calculated using Fused-GW.

## 4.4 Using GW-SDP on Realistic Shape-Matching Task

We use a publicly available dataset of triangular meshes (Sumner and Popović, 2004). The dataset comprises 72 objects from seven different classes, from which we chose samples of class cat, elephant, and horse. For each object, we first chose 4 representative points (the right back foot, the left front foot, the nose, and the tail) for each object and then selected another 14 points following the Euclidean farthest point sampling (fps) procedure. The distance matrices $C$ and $D$ are computed using Dijkstra's algorithm. Each object's probability measure is chosen to be uniform. We apply (GW-SDP) to the corresponding metric-measure spaces to determine the correspondence between the selected vertices across different objects. Two representative examples are given in Figure 5. For better visualization, in the representative examples we sampled only 6 points (4 representative points and 2 selected using fps).

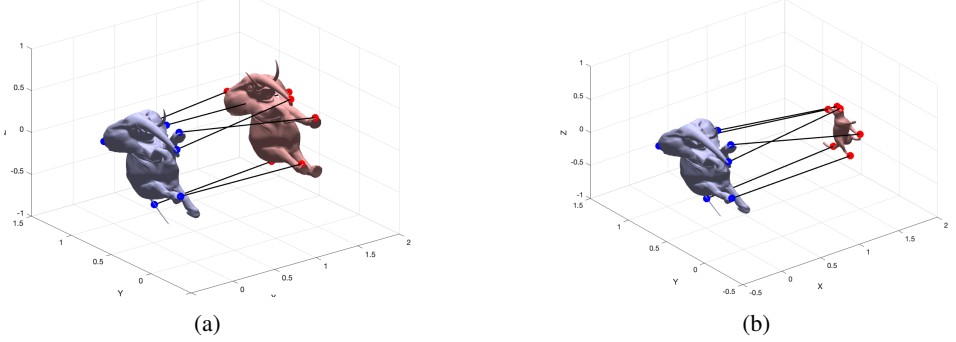

| (a) | (b) |

Figure 5: Correspondence between different 3D objects obtained by (GW-SDP). Left: Correspondence between two elephants. Right: Correspondence between an elephant and a cat. For both cases, (GW-SDP) returns one-one mappings.

Table 3 illustrates the results when we perform matching of distance matrices across different objects. In general, we expect shapes of the same animals to have a smaller GW distance than shapes of different animals, which is indeed the case for the three GW formulations. We still notice that GW-SDP consistently returns the smallest value when performing the same matching task.

Table 3: Value of different GW formulations for the realistic 3D shape matching dataset, visualized in Figure 5. GW-SDP consistently returns the smallest value when performing the same matching task.

|                   | GW-SDP   | GW-CG    | eGW-PPA  |
| ----------------- | -------- | -------- | -------- |
| Elephant-Elephant | 0.007416 | 0.043879 | 0.025688 |
| Elephant-Cat      | 0.015695 | 0.050594 | 0.042214 |
| Cat-Cat           | 0.006549 | 0.016634 | 0.006757 |
| Cat-Horse         | 0.011040 | 0.033736 | 0.011041 |
| Horse-Horse       | 0.006287 | 0.033768 | 0.007395 |

# 5 Duality

Given a generic optimization instance, the dual optimization instance concerns the task of finding optimal lower bounds to the primal instance. The (Lagrangian) dual to *any* optimization instance is a convex program in general and provides a principled way of obtaining convex relaxations of difficult optimization instances. We briefly discuss some of these relationships. A more detailed discussion on the duality can be found in Appendix B.

It is possible to obtain the relaxation (GW-SDP) via duality. Concretely, let (GW++) refer to the original GW problem instance (GW+) with the additional and *redundant* constraints (Nng.) and (Mar.). Let (GW-Dual) refer to the Lagrangian dual of the proposed semidefinite relaxation (GW-SDP).

**Theorem 5.1.** *The optimization instance* (GW-Dual) *is the Lagrangian dual of* (GW++)*, which is* (GW+) *with the additional constraints* (Nng.) *and* (Mar.)*.*

The (duality) gap between (GW-Dual) and (GW++) is non-zero in general, and is equal to zero precisely when the convex relaxation (6) succeeds. These can be characterized by a rank condition:

**Proposition 5.2.** *Let $P_{sdp}$ and $\pi_{sdp}$ be the solution to* (GW-SDP)*. Suppose the matrix variable has rank equals to one; that is*

$$rank \begin{pmatrix} P_{sdp} & vec(\pi_{sdp}) \\ vec(\pi_{sdp})^\top & 1 \end{pmatrix} = 1.$$

*Then the duality gap is zero; i.e., strong duality holds.*

# 6 Related work

There is a substantial body of prior work concerning the GW problem in the literature. We briefly discuss some of these and explain the novelty of our work.

First, the work in Vayer et al. (2019a) applies an alternating minimization-type approach based on the conditional gradient (Frank-Wolfe) algorithm to find local optima to the GW problem. This algorithm is currently implemented and is the default choice within the Python Optimal Transport package (Flamary et al., 2021). The basic idea is to start by computing the partial derivative of the objective (GW) with respect to $\pi$:

$$G(\pi) = 2\,\mathbf{L}(C, D) \otimes \pi,$$

This is a linear OT problem that can be solved using classical OT solvers. One proceeds with an alternating minimization scheme in which one updates the gradient $G$ with respect to $\pi^{(i-1)}$, subsequently solves for $\pi^{(i)}$ with the loss $G(\pi)$ at each $i$th-iteration, and finally projects $\pi^{(i)}$ into the feasible set by performing a line-search. The Conditional Gradient-based approach is not guaranteed to find globally optimal solutions; in fact, our numerical experiments in Section 4 suggest that this is quite often the case. Last, we briefly note that the work in Kerdoncuff et al. (2021) suggests a similar alternating numerical scheme.

Second, there is a body of work that aims at developing numerical schemes for finding transportation plans that approximately minimize the GW objective without incurring the expensive $O(m^2n^2)$ dependency. For instance, the work in Peyré et al. (2016) introduces an entropic regularization into the GW objective – this leads to a formulation that permits Sinkhorn scaling-like updates, much like the original scheme to solve entropic Wasserstein distance in Cuturi (2013). The work of Vayer et al. (2019b) adapts the ideas from the Wasserstein problem in one dimension in which closed-form solutions are available (this is known as the sliced Wasserstein problem, Rabin et al. 2012) to the GW context. Finally, the work in Sejourne et al. (2021); Vincent-Cuaz et al. (2021) relaxes the constraints on the probability distributions. These numerical schemes frequently lead to numerical schemes that are far more scalable than other existing methods, but they ultimately optimize for an objective that is different from the GW problem.

There is an interesting piece of work in Scetbon et al. (2022), which operates under the assumption that the cost matrices have low-rank structure. While the algorithm does not give guarantees about global optimality, it raises an interesting future direction; namely, could we develop numerical schemes for our proposed SDP relaxation that also exploit similar structures?

Finally, we discuss prior works that do in fact address global optimality (which is the heart of this paper): a recent work is in Mula and Nouy (2022), which suggests the use of moment sum-of-square (SOS) relaxation technique to solve the GW problem. The standard SDP relaxation for QPs on which our work is based on may be viewed, in a suitable sense, as the first level of the SOS hierarchy for polynomial optimization. Unfortunately, and as we note in Section 3, this alone is insufficient – the real novelty in our work is the addition of constraints that substantially strengthen the overall convex relaxation. A piece of related work by Villar et al. (2016) proposes a SDP relaxation of the closely related Gromov-Hausdorff problem, with an extension to the Gromov-Wasserstein problem. The relaxation is primarily designed for the Gromov-Hausdorff problem and is not equivalent to ours. The formulation also requires the probability distributions to be uniform whereas we do not. Another recent work by Ryner et al. (2023) also studies the Gromov-Hausdorff problem, and proposes a Branch-and-Bound approach for solving integer programs. The GW problem does not contain integer constraints, and hence Branch-and-Bound techniques are not applicable. That said, SDP relaxations can be used in conjunction with Branch-and-Bound. It would be interesting to see if our proposed SDP relaxations for GW suggest suitable relaxations for the Gromov-Hausdorff problem, which can be used in conjunction with the Branch-and-Bound techniques in Ryner et al. (2023).

## 7    Conclusions and Future Directions

In this work, we proposed a semidefinite programming relaxation of the Gromov-Wasserstein distance. Our initial results suggest that the relaxation (GW-SDP) is strong in the sense that $\pi_{sdp}$ frequently coincides with the globally optimal solution; moreover, we are able to provide a proof when this actually happens. These results are exciting, as it suggests a tractable approach for solving the GW problem – at least for examples of interest – which was previously assumed to be quite difficult.

An interesting future direction is to understand precisely how difficult is an instance of the GW problem. The fact that our convex relaxations work very well for the examples we considered suggests that the GW problem might not be as difficult as we think. It is important to bear in mind that these cost tensors $\mathbf{L}$ have structure – they arise from the difference of actual cost matrices. Could it be that the difficult instances of the GW problem correspond to cost tensors $\mathbf{L}$ that are not realizable as the difference of cost matrices; e.g., they violate the triangle inequality? A concrete question to this end is: Is the GW problem corresponding to cost tensors $\mathbf{L}$ arising in practical instances tractable to solve?

A second important future direction concerns computation. One limitation of our proposed convex relaxation is that it is specified as the solution of an SDP in which the matrix dimension is $mn$; that is, it is equal to the dimension of the transport plan. The prohibitive dependence on the data dimension means that we are currently only able to apply the relaxation on moderate sized instances using off-the-shelf SDP solvers. It would be of interest to develop specialized algorithms to solve the proposed relaxation (GW-SDP).

## Acknowledgements

We thank the anonymous reviewers for their helpful comments and feedback that helped improve our work. B.T. Nguyen is supported by NUS Start-up Grant A-0004595-00-00.

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

# A  Proofs of Main Results

*Proof of Proposition 3.1.* Let $1 \leq s, t \leq mn$ be coordinates such that $L_{st} > 0$. Let $v := v_{(s,t)} \in \mathbb{R}^{mn}$ be a vector whose $s$-th entry is 1, whose $t$-th entry is $-1$, and whose remaining entries are zeros. Let $\tilde{\pi} \in \Pi(\alpha, \beta)$ be any transportation map and consider the matrix $P_c = \mathrm{vec}(\tilde{\pi})\mathrm{vec}(\tilde{\pi})^\top + cvv^\top$. Notice that $P_c \succeq \mathrm{vec}(\tilde{\pi})\mathrm{vec}(\tilde{\pi})^\top$ for all $c \geq 0$. Hence the choice of variables $\pi = \tilde{\pi}$ and $P = P_c$ are feasible. Then notice that the objective evaluates to

$$\langle L, P_c \rangle = \langle L, \mathrm{vec}(\tilde{\pi})\mathrm{vec}(\tilde{\pi})^\top \rangle + c(L_{ss} - 2L_{st} + L_{tt}) = \langle L, \mathrm{vec}(\tilde{\pi})\mathrm{vec}(\tilde{\pi})^\top \rangle - 2cL_{st}.$$

We obtain the last equality by noting that $L_{ii} = 0$ for all $i$ (this is a property of cost matrices). The result follows by taking $c \to +\infty$. □

# B  Duality

Given a generic optimization instance, the dual optimization instance concerns the task of finding optimal lower bounds to the primal instance. The (Lagrangian) dual to *any* optimization instance is a convex program in general. As such, the process of deriving the dual to any optimization instance provides a principled way of obtaining convex relaxations of difficult optimization instances.

The objective value of the dual will always be a lower bound to the primal instance – this is precisely *weak duality*. In some cases, however, the objective values of these problems may coincide, and we call such settings *strong duality*.

In this section, we explore these relationships in the context of the GW problem. As we shall see, the proposed SDP relaxation (GW-SDP) can be viewed as being equivalent to the dual of an equivalent form of the GW problem (GW+), augmented with additional constraints that relate the linear and quadratic terms of transportation maps specified by (Nng.) and (Mar.). To simplify notation, we denote

$$a_i = \mathrm{vec}(e_i \mathbb{1}_n^\top),$$
$$b_j = \mathrm{vec}(\mathbb{1}_m e_j^\top).$$

We proceed to describe the dual program. We start by defining the following dual variables:

$$\begin{pmatrix} Y & y \\ y^\top & t \end{pmatrix} \succeq 0 \qquad\qquad : \begin{pmatrix} P & \mathrm{vec}(\pi) \\ \mathrm{vec}(\pi)^\top & 1 \end{pmatrix} \succeq 0 \tag{a}$$

$$\lambda_i \in \mathbb{R} \qquad\qquad : a_i^\top \mathrm{vec}(\pi) = \alpha_i, \quad i \in [m] \tag{b}$$

$$\mu_j \in \mathbb{R} \qquad\qquad : b_j^\top \mathrm{vec}(\pi) = \beta_j, \quad j \in [n] \tag{c}$$

$$Z \geq 0 \qquad\qquad : P \geq 0 \tag{d}$$

$$\eta^i \in \mathbb{R}^{mn} \qquad\qquad : Pa_i = \alpha_i \mathrm{vec}(\pi), \quad i \in [m] \tag{e}$$

$$\theta^j \in \mathbb{R}^{mn} \qquad\qquad : Pb_j = \beta_j \mathrm{vec}(\pi), \quad j \in [n] \tag{f}$$

As a reminder, the constraints in (b) and (c) specify the transportation map $\Pi(\alpha, \beta)$ while the constraints in (e) and (f) correspond to (Mar.).

**Theorem B.1.** *The Lagrangian dual of* (GW-SDP) *is given as follows*

$$\max \quad \lambda^\top \alpha + \mu^\top \beta - t$$

$$\text{s.t.} \quad \begin{pmatrix} L - Z + \frac{1}{2}\sum_{i=1}^m (\eta^i a_i^\top + a_i \eta^{i\top}) + \frac{1}{2}\sum_{j=1}^n (\theta^j b_j^\top + b_j \theta^{j\top}) & y \\ y^\top & t \end{pmatrix} \succeq 0,$$

$$\hspace{6cm} \text{(GW-Dual)}$$

$$\sum_{i=1}^m (\lambda_i a_i + \alpha_i \eta^i) + \sum_{j=1}^n (\mu_j b_j + \beta_j \theta^j) + 2y \leq 0,$$

$$Z \geq 0.$$

*Furthermore, strong duality holds; that is, the duality gap between* (GW-SDP) *and* (GW-Dual) *is zero.*

It turns out that it is possible to derive the dual instance (GW-Dual) *directly* from (GW+), an equivalent formulation of the original GW problem, with additional constraints specified by (Nng.) and (Mar.). Concretely, consider

$$
\begin{aligned}
\min_{\substack{\pi \in \mathbb{R}^{m \times n} \\ P \in \mathbb{R}^{mn \times mn}}} \quad & \langle L, P \rangle \\
\text{s.t.} \quad & P = \text{vec}(\pi)\text{vec}(\pi)^\top \\
& \pi \in \Pi(\alpha, \beta) \\
& P\text{vec}(e_i \mathbb{1}_n^\top) = \alpha_i \text{vec}(\pi), i \in [m] \\
& P\text{vec}(\mathbb{1}_m e_j^\top) = \beta_j \text{vec}(\pi), j \in [n] \\
& P \geq 0.
\end{aligned}
\qquad \text{(GW++)}
$$

The last three constraints on $P$ are always satisfied so long as $P = \text{vec}(\pi)\text{vec}(\pi)^\top$, where $\pi \in \Pi(\alpha, \beta)$ is a transportation map. Hence these constraints on $P$ and $\pi$ are technically redundant within (GW++). That is, the optimization instances (GW+) and (GW++) are equivalent. However, the Lagrangian dual of these optimization instances are different, and we summarize this observation in the following.

**Theorem B.2.** (GW-Dual) *is the Lagrangian dual of* (GW++).

The (duality) gap between (GW-Dual) and (GW++) is non-zero in general, and is equal to zero precisely when the convex relaxation (6) succeeds. These can be characterized by a rank condition satisfied by the optimal solutions, namely:

**Proposition B.3.** *Let $P_{sdp}$ and $\pi_{sdp}$ be the solution to GW-SDP. Suppose the matrix variable has rank equals to one, that is*

$$
\text{rank} \begin{pmatrix} P_{sdp} & \text{vec}(\pi_{sdp}) \\ \text{vec}(\pi_{sdp})^\top & 1 \end{pmatrix} = 1.
$$

*Then the duality gap for* (GW++) *is zero; i.e., strong duality holds.*

*Proof.* The rank condition implies $P_{sdp} = \text{vec}(\pi_{sdp})\text{vec}(\pi_{sdp})^\top$. Subsequently, the choice of variables $\pi = \pi_{sdp}$ and $P = \text{vec}(\pi_{sdp})\text{vec}(\pi_{sdp})^\top$ is a feasible solution to (GW++). This means (GW++) attains the same objective value as the (GW-Dual). Recall from Theorem B.2 that (GW-Dual) is the dual of (GW++), and hence in this instance the duality gap is indeed zero. $\qquad\square$

We summarize the relationships among the original GW problem, (GW+), (GW++), (GW-SDP), and (GW-Dual) in Figure 6:

- (GW+) is an equivalent reformulation of the original GW problem.
- (GW++) is derived by introducing additional redundant constraints to (GW+). Consequently, (GW+) and (GW++) share the same optimal solutions.
- (GW-SDP) is obtained by applying the standard SDP relaxation to (GW+) and introducing supplementary constraints to tighten the relaxation.
- (GW-Dual) serves as the Lagrangian dual to both (GW++) and (GW-SDP), and strong duality establishes the equivalence between (GW-Dual) and (GW-SDP).

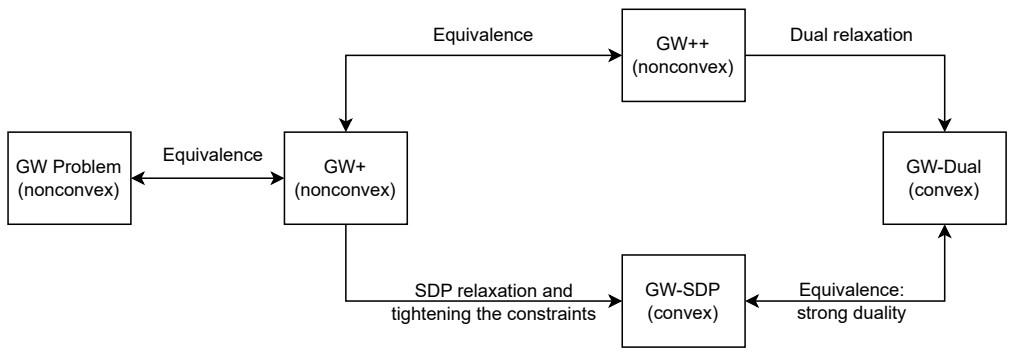

Figure 6: The relationship among the original GW problem, (GW+), (GW++), (GW-SDP) and (GW-Dual).

## C   Proof of Results Concerning Duality

We begin by defining these functions:

$$H(\eta, \theta, Z) := L - Z + \frac{1}{2}\sum_{i=1}^{m}(\eta^i a_i^\top + a_i \eta^{i^\top}) + \frac{1}{2}\sum_{j=1}^{n}(\theta^j b_j^\top + b_j \theta^{j^\top}),$$

$$g(\lambda, \mu, \eta, \theta, y) := \sum_{i=1}^{m}\left(\lambda_i a_i + \alpha_i \eta^i\right) + \sum_{j=1}^{n}\left(\mu_j b_j + \beta_j \theta^j\right) + 2y.$$

*Proof of Theorem B.1.* [Deriving the dual program]: The dual function of the GW-SDP problem is given by

$$\min_{P, \pi \geq 0}\ \operatorname{tr}(LP) - \operatorname{tr}(YP) - 2y^\top \operatorname{vec}(\pi) - t$$

$$+ \sum_{i=1}^{m}\lambda_i(\alpha_i - a_i^\top \operatorname{vec}(\pi)) + \sum_{j=1}^{n}\mu_j(\beta_j - b_j^\top \operatorname{vec}(\pi))$$

$$+ \sum_{i=1}^{m}\eta^{i^\top}\left(Pa_i - \alpha_i \operatorname{vec}(\pi)\right) + \sum_{j=1}^{n}\theta^{j^\top}\left(Pb_j - \beta_j \operatorname{vec}(\pi)\right)$$

$$- \operatorname{tr}(ZP)$$

$$= \min_{P}\ \operatorname{tr}\left((H(\eta, \theta, Z) - Y)P\right)\ +\ \min_{\pi \geq 0}\ \langle -g(\lambda, \mu, \eta, \theta, y), \operatorname{vec}(\pi)\rangle$$

$$+ \lambda^\top \alpha + \mu^\top \beta - t.$$

In the above minimization over $P$, we observe that the objective evaluates to $-\infty$ if the following does not hold

$$H(\eta, \theta, Z) = Y.$$

Similarly, in the minimization over $\pi \geq 0$, the objective evaluates to $-\infty$ if the following does not hold

$$g(\lambda, \mu, \eta, \theta, y) \leq 0.$$

We impose these as constraints, and we add the additional constraint that $Y \succeq 0$ on our dual variable, to obtain the form of the dual problem in (GW-Dual).

[Establishing zero duality gap]: Notice that (GW-SDP) and (GW-Dual) are convex programs. Hence, to show strong duality, it suffices to check that Slater's condition hold; that is, there exists a strictly feasible solution.

Consider $\eta^i = (|\lambda_{\min}(L)| + 2)\mathbb{1}$, $\lambda_i = -2m$ for $i \in [m]$, $\theta^j = 0$, $\mu_j = 0$ for $j \in [n]$, $t = 1$, $y = 0$, and

$$Z = (|\lambda_{\min}(L)| + 2)\mathbb{1}\mathbb{1}^\top - (|\lambda_{\min}(L)| + 1)I$$

for some $\mathbb{1}$, $0$ and $I$ of appropriate dimension. Then we have $Z > 0$, and

$$g(\lambda, \mu, \eta, \theta, y) = -2m \sum_{i=1}^{m} a_i + \sum_{i=1}^{m} \mathbb{1} = -m\mathbb{1} < 0.$$

Additionally, since $H(\eta, \theta, Z) = L + (|\lambda_{\min}(L)| + 1)I \succ 0$, $t > 0$ and $y = 0$, it follows that the LHS of the first constraint in (GW-Dual) is positive definite. Therefore, we find a feasible solution of (GW-Dual) such that strict inequality holds for all inequality constraints. Strong duality then follows. $\qquad\square$

*Proof of Theorem B.2.* In addition to the dual variables (b)-(f), we define these additional dual variables:

$$Y \in \mathbb{R}^{mn \times mn} \quad : P = \text{vec}(\pi)\text{vec}(\pi)^\top$$
$$z \geq 0 \quad\quad\quad : \text{vec}(\pi) \geq 0$$

Then the dual function of (GW++) is given by

$$\min_{\pi, P} \text{tr}(LP) - \text{tr}(Y(P - \text{vec}(\pi)\text{vec}(\pi)^\top)) + \sum_{i=1}^{m} \lambda_i(\alpha_i - a_i^\top \text{vec}(\pi)) + \sum_{j=1}^{n} \mu_j(\beta_j - b_j^\top \text{vec}(\pi))$$

$$+ \sum_{i=1}^{m} \eta^{i\top}(Pa_i - \alpha_i\text{vec}(\pi)) + \sum_{j=1}^{n} \theta^{j\top}(Pb_j - \beta_j\text{vec}(\pi)) - \text{tr}(ZP) - z^\top\text{vec}(\pi)$$

$$= \min_P \underbrace{\text{tr}((H(\eta, \theta, Z) - Y)P)}_{A_1}$$

$$+ \min_\pi \underbrace{\text{vec}(\pi)^\top Y \text{vec}(\pi) - \left(\sum_{i=1}^{m}(\lambda_i a_i + \alpha_i\eta^i) + \sum_{j=1}^{n}(\mu_j b_j + \beta_j\theta^j) + z\right)^\top \text{vec}(\pi)}_{A_2}$$

$$+ \lambda^\top\alpha + \mu^\top\beta.$$

To simplify notation, we denote

$$p := \sum_{i=1}^{m}(\lambda_i a_i + \alpha_i\eta^i) + \sum_{j=1}^{n}(\mu_j b_j + \beta_j\theta^j).$$

Observe that

$$\min_{P \in \mathbb{R}^{mn \times mn}} A_1 = \begin{cases} 0 & , \quad \text{if } Y = H(\eta, \theta, Z) \\ -\infty & , \quad \text{otherwise} \end{cases},$$

and

$$\max_{\pi \in \mathbb{R}^{m \times n}} A_2 = \begin{cases} -\frac{1}{4}(p+z)^\top Y^\dagger(p+z) & , \quad \text{if } Y \succeq 0 \text{ and } (I - YY^\dagger)(p+z) = 0 \\ -\infty & , \quad \text{otherwise} \end{cases}$$

Hence, the dual of (GW++) is given by

$$\max_{\lambda, \mu, y, z, Z} \quad \lambda^\top\alpha + \mu^\top\beta - \frac{1}{4}(p+z)^\top Y^\dagger(p+z)$$
$$\text{s.t.} \quad Y \succeq 0$$
$$\quad (I - YY^\dagger)(p+z) = 0 \quad\quad\quad .$$
$$\quad Y = H(\eta, \theta, Z)$$
$$\quad Z \geq 0$$
$$\quad z \geq 0$$

We re-write this as

$$\max_{\lambda,\mu,y,z,Z,t} \quad \lambda^\top \alpha + \mu^\top \beta - t$$

$$\text{s.t.} \quad \frac{1}{4}(p+z)^\top Y^\dagger (p+z) \le t$$
$$Y \succeq 0$$
$$(I - YY^\dagger)(p+z) = 0$$
$$Y = H(\eta, \theta, Z)$$
$$Z \ge 0$$
$$z \ge 0$$

By taking Schur complements and by replacing $Y$ with $H(\eta, \theta, Z)$, the above optimization instance reduces to

$$\max_{\lambda,\mu,z,Z,t} \quad \lambda^\top \alpha + \mu^\top \beta - t$$

$$\text{s.t.} \quad \begin{pmatrix} H(\eta, \theta, Z) & -\frac{1}{2}(p+z) \\ -\frac{1}{2}(p+z)^\top & t \end{pmatrix} \succeq 0$$
$$Z \ge 0$$
$$z \ge 0$$

Note that $g(\lambda, \mu, \eta, \theta, y) = p + 2y$, the theorem then follows by doing a change of variable $y = -\frac{1}{2}(p+z)$. $\qquad \square$

# D  Extended Applications of GW-SDP

## D.1  GW-SDP Barycenters

One popular application of optimal transport is to compute the barycenters of measures that serves as a building block for many learning methods. The notion of barycenter for measures was first proposed in Agueh and Carlier (2011) for Wasserstein space. Akin to barycenter in Euclidean space (Fréchet), the Wasserstein barycenter is defined as the solution of a weighted sum of OT distances over the space of measures. An efficient algorithm to compute the discrete OT barycenter with entropic regularization was proposed in Benamou et al. (2015), and was later extended to discrete metric-measure spaces with entropic GW distance in Peyré et al. (2016).

We show that it is straightforward to extend the GW-SDP formulation to find barycenters of a set of data as Fréchet means. For simplicity, we assume that the base histogram $\bar{\alpha}$, the size of the barycenters $m \in \mathbb{N}$, and $(\lambda_k)_k$ such that $\sum_k \lambda_k = 1$ are fixed. We aim to find a structure matrix $\bar{C}$ that minimizes

$$\min \sum_k \lambda_k \text{GW-SDP}(C_k, \bar{C}, \alpha_k, \bar{\alpha}). \tag{7}$$

We have the following corollary.

**Corollary D.1** (Adaptation of Proposition 3 in Peyré et al. (2016)). *In the special case of the squared loss $\ell(a, b) = (a - b)^2$, the solution of (7) reads*

$$\bar{C} = \frac{\sum_k \lambda_k \pi_{sdp,k}^\top C_k \pi_{sdp,k}}{\alpha \alpha^\top}, \tag{8}$$

*where $\pi_{sdp,k}$ is the solution to GW-SDP$(C_k, \bar{C}, \alpha_k, \bar{\alpha})$ and the division is entry-wise.*

Corollary D.1 shows that we may apply iterative updates to solve for the barycenter $\bar{C}$ via the Block Coordinate Descent (BCD) algorithm. At each iteration, we solve $K$ independent instances of the GW-SDP problem to find $(\pi_{sdp,k})_k$, and then compute $\bar{C}$ using (8) to solve for (7). A pseudocode for the GW-SDP barycenter calculation is provided in Algorithm 1. We demonstrate the effectiveness of the GW-SDP barycenter calculation by applying it to find the barycenter of a graph dataset. The dataset consists of 20 noisy graphs, created by adding random connections from a circular graph. We show a visualization of 9 of these in Figure 7a. The number of nodes ranges from 8-16. We apply the (GW-SDP) barycenters update for 100 iterations, and Figure 7b shows the result for a circular graph of 10 nodes.

---
**Algorithm 1** Computation of GW-SDP barycenters.
---

**Input:** dataset $\{C_k, \alpha_k\}_{k=1}^K$; $\{\lambda_k\}_{k=1}^K$.
Initialize $\bar{C}$.
**repeat**
  **for** $k = 1$ **to** $K$ **do**
    $\pi_{sdp,k} \leftarrow \texttt{solve\_GW-SDP}(C_k, \bar{C}, \alpha_k, \bar{\alpha})$.
  **end for**
  Update $\bar{C}$ using (8).
**until** convergence

---

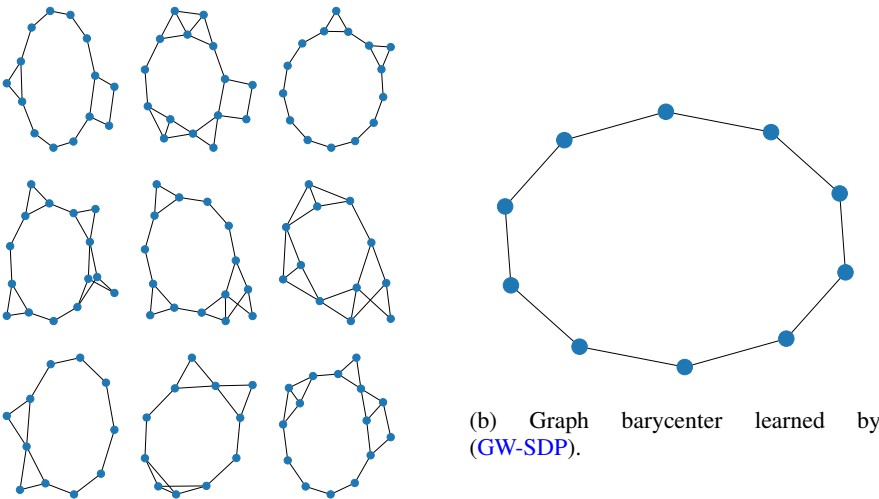

(a) Visualization of noisy circular graphs.

(b) Graph barycenter learned by (GW-SDP).

Figure 7: Application of the (GW-SDP) to find graph barycenter of noisy circular graphs.

## D.2   Outlier-Robust GW-SDP

It is generally possible to extend the semidefinite relaxation to variants of the GW problem. We briefly describe the semidefinite relaxation to the outlier-robust GW problem by Kong et al. (2024). Here, $(X, d_X)$ and $(Y, d_Y)$ are two metric spaces with accompanying measures $\mu$ and $\nu$. The distance between $\mu$ and $\nu$ is

$$\min \ \langle L, P \rangle + \tau_1 d_{KL}(\pi 1, \alpha) + \tau_2 d_{KL}(\pi^T 1, \beta)$$

$$\text{s.t.} \begin{pmatrix} P & \text{vec}(\pi)^T \\ \text{vec}(\pi) & 1 \end{pmatrix} \succeq 0$$

$$\sum_i P_{(i,j),(k,l)} = f_j^{k,l}, \Sigma_j P_{(i,j),(k,l)} = g_i^{k,l}$$

$$P \geq 0$$

$$d_{KL}(\mu, \alpha) \leq \rho_1, d_{KL}(\nu, \beta) \leq \rho_2.$$

There is one technical aspect: In the marginal sums $\sum_i P_{(i,j),(k,l)}$ we set this equal to some constant $f_j^{k,l}$. In GW-SDP, the corresponding RHS term depends on $\pi$ and $\alpha$. In the robust set-up, $\alpha$ is an optimization variable, not a constant, which necessitates the above change. We remark that the resulting formulation is convex but not an SDP because of the presence of the KL divergence.

