# OpenReview forum: "Semidefinite Relaxations of the Gromov-Wasserstein Distance"
_NeurIPS.cc/2024/Conference — NeurIPS 2024 poster_

### Official Review · Reviewer_Xfoj · 2024-07-06

**Soundness:** 3
**Presentation:** 3
**Contribution:** 2
**Rating:** 6
**Confidence:** 4

**Summary:**

The paper explores a semi-definite programming (SDP) based relaxation of the popular Gromov Wasserstein (GW) problem. The GW problem  is an instance of non-convex quadratic program (QP). Standard SDP relaxation of QPs has been explored in the literature. The present work leverages this SDP relaxation result. However, this standard SPD relaxation is not sufficient as the resulting minimization problem is unbounded from below. Hence, the paper tightens the relaxation via additional constraints which are motivated from the GW problem. Empirical evaluations are performed to showcase the effectiveness of the obtained solution, both in terms of quality and runtime efficiency.

**Strengths:**

- The proposed SDP relaxation based approach for GW problem is an interesting idea and has not been explored in context of GW (to the best of my knowledge). This reformulation has interesting consequences such as
    - an approximation ratio (Eqn 4) which can be computed from the solution obtained via the proposed solution. This ratio lower bounded by 1 and is equal to 1 only if the obtained solution is globally optimal. Hence, this approach provides an optimality certificate for a non-convex problem
    - While no empirical results were shown in this regard, the proposed approach allows using a general GW cost tensor L. Existing approach such as (Peyré et al., 2016) can only employ decomposable costs (such as those obtained via L2 or KL loss).

- The paper empirically evaluates the effectiveness of the proposed approach in terms of computational efficiency and quality (lower objective is better). While the proposed approach obtains better objective compared to current state-of-the-art GW-CG, its runtime (for problems of size n = 6,12,20) is around 500-150000 times higher than GW-CG (Table 1).  While the heuristic solver (GW-PGD) proposed in Section 5 is faster than the proposed GW-SDP, it is still at least 250 times slower than GW-CG and its objective becomes comparable to GW-CG as n increase.

- The paper is well written, explaining the underlying concepts and the related works nicely.

**Weaknesses:**

- The paper provides a detailed discussion on the literature related to the quadratic assignment problem (QAP) and SDP relaxations of the QPs. The key technical contribution of the proposed work w.r.t. Zhao et al. (1998) is removing constraints related to \pi being a permutation matrix, which also allows handling the m \neq n setting (lines 125-137).

- Very high runtime compared to GW-CG. This limits the practical utility of proposed GW-SDP or GW-PGD.

**Questions:**

please see strength and weakness sections.

---

> ### Author Rebuttal · Authors · 2024-08-07
>
> *Detailed discussion on related work*
>
> Could we check if this comment was intended as a weakness? It does not sound like one. We assume this comment was misplaced. Could the Reviewer clarify?
>
> *High run-time*
>
> We agree with the Reviewer’s concerns about run-time. We address some of these concerns in the global response.
> We have been working on developing faster algorithms for solving the GW-SDP. Unfortunately, this is a difficult task. There are many promising directions to develop more scalable algorithms for solving the GW-SDP problem. Some of these were discussed in the manuscript. We discuss first-order methods in the response to another Reviewer. These directions are promising but the scope of the work goes comfortably the scope of a single paper.
>
> As a note, for the closely related semidefinite relaxation of the QAP, state-of-the-art methods are only able to solve problem instances where the dimension is around 40 [OWX:18]. The largest dimension in our experiments is 32, which is not far off the state of the art.
>
> *References*
>
> [OWX:18] D. E. Oliveira, H. Wolkowicz, & Y. Xu, (2018). ADMM for the SDP relaxation of the QAP. Mathematical Programming Computation, 10(4), 631-658.

---

> > ### Comment · Reviewer_Xfoj · 2024-08-10
> > **Response to authors' rebuttal**
> >
> > I thank the authors for their response. I also went through other reviews and their discussions. Please find my observations below:
> >
> > 1. The detailed discussion on QAP related work is not a weakness and is much appreciated. However, from solution approach point of view, the key technical contribution of this works seems to be only the removal of constraints related to \pi being a permutation matrix, which also allows handling the m \neq n setting (lines 125-137). And this was done because GW problem setting is interesting also for m \neq n setting. Hence, novelty seems limited (and this was the weakness part, perhaps my earlier statement was not very clear). Regarding novelty, please also see the next point.
> >
> > 2. The approximation ratio (Eqn 4) is also a novel contribution and interesting. However, in their global comments and one of the rebuttals to another review, the authors mention statements such as "... our work is the only work we are aware of that addresses global optimality ... " and "The first contribution is that we compute globally optimal solutions to the GW problem. We can prove that our solutions are globally optimal." I think this is a gross exaggeration of their contribution. My understanding of the contribution of this paper in this regard is as follows. Please excuse the verbosity.
> > - The paper *does not* propose an algorithm that is guaranteed to converge to a globally optimal solution of the (discrete) GW problem.
> > - The paper *can only confirm* whether the solution obtained by *its approach* has converged to a globally optimal solution.
> > - The paper *has empirically shown* that in several (very) small scale problem setting, it achieves global optimality. Whether similar observations would hold for larger setting is unclear.
> > - However, given a problem instance, we do not know beforehand whether the proposed approach will converge sub-optimally or otherwise.
> > - Given just a solution to GW problem, the paper's approximation ratio cannot be used to check whether the given solution is optimal or not.
> > - In the (limited) additional experiments during rebuttal phase, the paper has also shown empirically that in the experiment that were performed, their approach reach optimal solution when m is a multiple of n. However, no theoretical justification is provided.
> >
> > 3. In the additional experiments done with more sample points for FW, the setting seems quite synthetic since the source and target points are sampled from a distribution (with no noise). In practice, the source and target datasets could be corrupted by noise/outliers. Say the source and target datasets had 5000 datapoints with some outliers as well. GW-SDP maybe run by sampling 10 points from source and target datasets. Does GW-SDP seem more susceptible to outliers since only 10 points were taken and hence an outlier has a greater chance to influence the results? In certain applications where the GW transport map required for domain adaptation [1], such sampling would again not be useful.
> >
> > [1]  Gromov-Wasserstein Alignment of Word Embedding Spaces. EMNLP 2018.
> >
> > 4. While the authors have acknowledged and discussed it in the draft as well as in their response, the concerns about run-time and scalability of the approach, and its practical relevance remain.

---

> ### Author Response · Authors · 2024-08-10
> **Response to comment by Reviewer Xfoj**
>
> When we wrote that "... our work is the only work we are aware of that addresses global optimality ... " and "The first contribution is that we compute globally optimal solutions to the GW problem. We can prove that our solutions are globally optimal.", it was to be understood in the context that global optimality was attained if the computed ratio was one.
>
> When writing the rebuttal, we believed that all Reviewers understood the conditions under which global optimality held. Hence in writing a rebuttal, when summarizing our contributions, we did not think it was necessary to repeat the conditions under which global optimality held.
>
> Unfortunately, this summary has given the impression that we claimed global optimality in all instances. We do not claim such a thing. We hope it was clear from the original paper we do not claim global optimality in all instances. We apologize for the confusion. This was not our intention. We hope the paper was very clear about its claims and limitations.
>
> > Given just a solution to GW problem, the paper's approximation ratio cannot be used to check whether the given solution is optimal or not.
>
> To check if a solution is optimal, one typically needs to compute the dual problem (of some suitable form). Computing the dual is (sort of) unavoidable if you want a proof of optimality. The GW-SDP does this (in a way) because it is a convex program.
>
> Also, given the solution to the GW-SDP formulation, one can certify the optimality, or the gap to optimality, of any proposed solution to the GW problem. Compute the approximation ratio. If it is equal to one, the proposed solution is optimal. If it is close to one, we know it is quite good.
>
> > In the (limited) additional experiments during the rebuttal phase, the paper has also shown empirically that in the experiment that were performed, their approach reach optimal solution when m is a multiple of n. However, no theoretical justification is provided.
>
> We do not provide theoretical justification. In fact, we should not expect the SDP to provide exact solutions to all instances because the GW problem is not known to be tractable to solve. There must be some inputs for which the SDP is not exact. The surprising aspect here is that the SDP is exact for many inputs, and that is the point our experiments make.
>
> (As to providing theoretical justification, such as proving such a phenomenon holds for random instances, is a difficult research problem. We don't have an answer for this at the moment.)
>
> The Reviewer posed some further questions about outliers. We need a bit more time to respond to this and will provide a reply later.

---

### Official Review · Reviewer_fpsx · 2024-07-08

**Soundness:** 2
**Presentation:** 3
**Contribution:** 1
**Rating:** 3
**Confidence:** 4

**Summary:**

The authors propose a new algorithm for measuring the Gromov-Wasserstein (GW) distance, an metric for assessing the similarity of point clouds in different spaces. Their algorithm formulates the computation of the GW distance as a quadratic programming problem, which is then solved by semidefinite relaxation. The authors conducted experiments using several synthetic datasets and confirmed that the proposed algorithm yields solutions with smaller objective function values compared to other algorithms.

**Strengths:**

* The proposed method is grounded in solid theoretical foundations.
* The proposed method has been experimentally verified to produce solutions with smaller objective function values.
* The paper is well-written, and its contributions are clearly described.

**Weaknesses:**

* The novelty is limited. It is well-known that the computation of GW can be reduced to a quadratic programming (QP) problem, and solving a non-convex QP through semi-definite relaxation is a very common approach in the field of optimization. Additionally, as the authors themselves point out, such methods have been proposed for the quadratic assignment problem (QAP), which is closely related to GW. Thus, the proposed method is merely a simple variant of these approaches, and its technical contribution is minimal.

* The computational complexity of the proposed method. The proposed method requires solving an SDP with a matrix of size
$mn \times mn$ as variables. Although SDPs can indeed yield globally optimal solutions with relatively low computational effort, it is well-known that the computational complexity increases sharply with the size of the problem. It is thus challenging to solve an SDP with a matrix of size $mn \times mn$ as variables in practical applications. In fact, Table 1 shows that for
$n=20$, the computation time exceeds 200 seconds, indicating difficulties in applying the method to real-world problems.

* Insufficiency of the experiments. The experiments conducted in the paper are all small-scale and limited to ten artificial datasets, which is insufficient to demonstrate the effectiveness of the proposed method.

**Questions:**

* The experiments emphasize the sparsity of the output of the proposed method, but what is the benefit of this sparsity? Additionally, why does sparsity emerge in the results?

* The positioning of the Heuristic Solver proposed in Section 5 within this paper is unclear. Is this method one of the proposals of this paper? If so, it is necessary to explain its position within existing research and its motivation in more detail. If not, its contribution becomes unclear, and it is recommended to move it to the appendix.

**Limitations:**

The authors are aware of the significant computational complexity and discuss potential directions to address this issue. While this acknowledgment is commendable, the substantial computational complexity remains a critical drawback of this method. Unless this issue is resolved, the overall contribution of the paper must be considered limited.

---

> ### Author Rebuttal · Authors · 2024-08-07
>
> *Limited novelty*
>
> We emphasize, the paper has multiple contributions. The first contribution is that we compute globally optimal solutions to the GW problem. We can prove that our solutions are globally optimal. There is no existing work (as far as we are aware of), published or unpublished, that achieves this. This contribution alone, we believe, is a significant contribution.
>
> The second contribution is to point out that many existing methods do not compute globally optimal solutions. This point, we believe, is a very concerning development, and researchers working on GW need to be aware.
>
> The Reviewer is concerned with the limited novelty because the relaxation is a variant of an existing relaxation for the QAP. The point we make here is that this particular variant we propose is the simplest variant that answers the previous two questions.
>
> We hope the Reviewer appreciates the conceptual aspects of our paper, namely that global optimality and certificate of optimality, are fundamental aspects of optimization.
>
> *Sparsity*
>
> When $m=n$, the feasible set of the GW problem is the set of doubly stochastic matrices. The extreme points of the set of doubly stochastic matrices is the collection of permutation matrices – this is known as the Birkhoff von-Neumann theorem. When $m \neq n$, the feasible set is known as the transportation polytope. The extreme points correspond to bipartite graphs (the set of vertices are $\alpha$ and $\beta$) that have no cycles. These matrices are also sparse.
>
> The optimal solution of a convex program whose feasible region is the set of doubly stochastic matrices or more generally the transportation polytope tends to be sparse. This is because optimal solutions tend to lie on low-dimensional faces, which are sparse (this is a known phenomenon in convex geometry).
>
> In short, the optimal solution to the GW problem should generally be sparse. The solutions to GW-SDP are sparse, and this is a good sign. The solutions obtained by Conjugate Gradient tend to be sparse, which is also a good sign. On the other hand, if the solution is non-zero everywhere, we can be certain the solution is sub-optimal. This is the case for e-GW solvers, and the reason is that e-GW adds a bias to the objective; that is, they solve a slightly different problem.
>
> *Heuristic solver*
>
> We will place the heuristic solver in the appendix, as recommended.
>
> *Unless this issue is resolved, the overall contribution of the paper must be considered limited.*
>
> We hope the Reviewer will take into consideration that developing scalable algorithms for solving large-scale SDPs is an active research area and an open challenge. In fact, for the closely related semidefinite relaxation of the QAP, state-of-the-art methods are only able to solve problem instances where the dimension is around 40 [OWX:18]. The largest dimension in our experiments is 32, which is not far off the state of the art.
>
> There are many promising directions to develop more scalable algorithms for solving the GW-SDP problem. Some of these were discussed in the manuscript. We discuss first-order methods in the response to another Reviewer. These directions are promising but the scope of the work goes comfortably the scope of a single paper. We urge the Reviewer to moderate expectations as to what is technically feasible within the scope of a single paper.
>
> Again, we reiterate an earlier point, that there is no other work we are aware of that addresses the global optimality of the GW problem. Addressing global optimality is one of the key contributions of this paper.
>
> *References*
>
> [OWX:18] D. E. Oliveira, H. Wolkowicz, & Y. Xu, (2018). ADMM for the SDP relaxation of the QAP. Mathematical Programming Computation, 10(4), 631-658.

---

> > ### Comment · Reviewer_fpsx · 2024-08-09
> > **Comments on the Authors' Rebuttal**
> >
> > I have read the authors' rebuttal and would like to address the following points:
> >
> > * Originality: The statement in the rebuttal, "The first contribution is that we compute globally optimal solutions to the GW problem," is inaccurate. To be precise, a solution is guaranteed to be globally optimal only when the right-hand side of equation (4) equals 1. Although Figure 4(b) shows that this condition is met in many cases, it is likely due to the small and simple problem setting. Therefore, the algorithm does not always yield a globally optimal solution, making the authors' claim in the rebuttal an overstatement. Moreover, as mentioned in my initial review, similar algorithms exist in the context of the Quadratic Assignment Problem (QAP).
> >
> > * Computation Time: My concerns regarding computation time remain unresolved. While I recognize the value of this research even without significant reductions in computation time, the impact of the work is diminished as a result.
> >
> > * Additional Experiments: I appreciate the additional experiments conducted by the authors.
> >
> > * Sparsity: My concerns regarding sparsity have not been fully addressed. While it is correct that an optimal solution should be sparse, the argument that a sparse solution is close to the optimal one is not valid. Therefore, I find little merit in discussing the sparsity of the solutions.
> >
> > Overall, my concerns have not been sufficiently resolved, and thus, I will maintain my previous score.

---

> ### Author Response · Authors · 2024-08-09
>
> We thank the reviewer for their quick response to our rebuttals.
>
> > Originality: The statement in the rebuttal, "The first contribution is that we compute globally optimal solutions to the GW problem," is inaccurate. To be precise, a solution is guaranteed to be globally optimal only when the right-hand side of equation (4) equals 1.
>
> In our paper, we do make it clear that we only claim global optimality when the RHS of equation (4) equals one.
>
> > Sparsity: My concerns regarding sparsity have not been fully addressed. While it is correct that an optimal solution should be sparse, the argument that a sparse solution is close to the optimal one is not valid. Therefore, I find little merit in discussing the sparsity of the solutions.
>
> The Reviewer asked why do solutions tend to be sparse. The initial comments do not appear to be a concern and we answered the question as it is. Our response simply states that solutions that are not sparse are certainly not optimal. Our proof of global optimality comes from the RHS of equation (4), not sparsity.

---

> > ### Author Response · Authors · 2024-08-10
> > **Addendum about global optimality claims**
> >
> > In reading the response by other reviewers, it appears that other reviewers share similar concerns about overstating the claims.
> >
> > For what it's worth, in writing the rebuttal, we intended to give a quick summary of our contributions before proceeding to the additional experiments we ran.
> >
> > In writing "The first contribution is that we compute globally optimal solutions to the GW problem," we intended this sentence to be understood that it holds under the condition the approximation ratio is equals to one. From the reading the reviews, we believed the reviewers understood the extent of our claims. As such, it did not occur to us to repeat the conditions (namely that these claims hold if the approximation ratio equals one, which seems to hold fairly often empirically).
> >
> > Unfortunately, it gave the inadvertent impression that we overstate our contributions.
> >
> > That is not our intention. We apologize for the inaccurate statement about our claims. We hope it was clear from our paper what the extent of our claims of global optimality were.

---

> > ### Comment · Reviewer_fpsx · 2024-08-11
> > **Comments on the Authors' Rebuttal**
> >
> > > For what it's worth, in writing the rebuttal, we intended to give a quick summary of our contributions before proceeding to the additional experiments we ran.
> >
> > I understand your intention. The original paper presents an accurate description, so I don't see issues with it. However, the rebuttal lacks persuasive power, and my concerns regarding the novelty remain unresolved.
> >
> > > The initial comments do not appear to be a concern and we answered the question as it is.
> >
> > The issue related to sparsity is not a major concern. However, I believe that the discussion on sparsity does not hold much significance and only serves to unnecessarily confuse the reader. Therefore, I suggest either removing the description or providing a more detailed explanation of its intent.

---

> > > ### Author Response · Authors · 2024-08-12
> > >
> > > >The issue related to sparsity is not a major concern. However, I believe that the discussion on sparsity does not hold much significance and only serves to unnecessarily confuse the reader. Therefore, I suggest either removing the description or providing a more detailed explanation of its intent.
> > >
> > > This is a useful suggestion and we will provide a more detailed explanation in the revision. Thanks for the suggestion.
> > >
> > > >I understand your intention. The original paper presents an accurate description, so I don't see issues with it. However, the rebuttal lacks persuasive power, and my concerns regarding the novelty remain unresolved.
> > >
> > > Thanks for understanding our intention, and we apologize for the confusion.
> > >
> > > As for novelty, let's perhaps try to state our case a little differently. Besides our work, we are not aware of other work that address the issue of global optimality. When we say "addresses global optimality", we mean that the method computes the actual GW distance (up to numerical error) with a proof that it does so correctly. Methods that depend on performing descent within a neighborhood do not achieve that. It may compute the global minimum, or it might get stuck in a local optimum, but it won't know which case it belongs in, with certainty. The typical way to certify that one has the global minimum is to compute a lower bound to the GW problem -- usually, this is done via a suitable dual program.
> > >
> > > The GW problem is believed to be computationally difficult (intractable). So we expect the dual problem to be equally difficult. Here, what we are offering is slightly different: We suggest a tractable (meaning polynomial time) method for computing a relaxation of the dual that is able to certify global optimality on many instances. Because we think finding the global optimal solution should be difficult (intractable), coming up with a tractable method that certifies global optimality in many instances is the next best thing one can hope for. This is precisely what we are offering.
> > >
> > > It is correct that the specific SDP formulation is a minor variation of a closely related SDP relaxation for the QAP. We were upfront about this in the manuscript as well. There are a few things to note: We considered a few other relaxations but they did not work because for many instances, the approximation ratio was vacuous. The SDP formulation we arrived at was a specific formulation that gave meaningful ratios most of the time. But the point here is not that our relaxation is particularly novel (and we were careful to attribute our ideas to the source); the point here is that the relaxation helps to certify global optimality in many instances. This we think is an important contribution. There are no other methods we know of in the literature that solves this seemingly basic problem.
> > >
> > > As such, the burden of the paper now shifts to quantifying how often the relaxation is exact. If exactness happens quite often, then the relaxation is useful. If exactness only occurs for very specific examples, then there is very little meaning to the relaxation. Our numerical experiments are intended to investigate how often the SDP is exact, and over fairly generic inputs. The largest instance we solve has $30$ points. In other words, the SDP is of size $900 \times 900$. This is considered moderate-sized for SDP. Most researchers working with SDPs would consider such a size very promising (meaning it suggests the relaxation is a powerful one). Now, $30$ points is considered small in data science and optimal transport applications. But the more relevant question here is the strength of the SDP, perhaps more so than the data analytical aspects.  Now, the Reviewer suggests that the exactness is due to the "small and simple" settings.  We don't agree with this characterization and our experiments are intended to suggest otherwise.  If the Reviewer has further reasoning to explain why the Reviewer thinks so, do let us know and we will be happy to engage.
> > >
> > > In any case, thank you very much for your attention.

---

### Official Review · Reviewer_3Vyi · 2024-07-11

**Soundness:** 3
**Presentation:** 3
**Contribution:** 3
**Rating:** 6
**Confidence:** 4

**Summary:**

The authors propose a semidefinite programming (SDP) relaxation of the Gromov-Wasserstein (GW) distance. While the GW problem is non-convex, the proposed SDP relaxation is convex and hence can be solved in polynomial time with any off-the-shelf convex solver. The authors also provide an accompanying proof of global optimality for the relaxed problem, which can be checked efficiently. The numerical experiments use an off-the-shelf solver to solve the proposed SDP relaxation and compare it with two solvers from the PythonOT package, i.e.,  Conditional Gradient (CG-GW) solver and  Sinkhorn projections solver on Matching Gaussian Distributions and Graph Community Matching. Lastly, the authors present a simple heuristic algorithm optimized for their proposed SDP relaxation.

**Strengths:**

* The authors propose a novel SDP relaxation of the GW problem, which has not previously been explored in the literature. The theory is compelling.
* The proposed SDP approach has several compelling advantages over existing GW solvers: the solver given by POT (which implements Frank-Wolfe) can only find local optima, and entropic GW cannot be used for general cost tensors. Meanwhile, the proposed approach is broadly applicable and to my knowledge the first tractable GW solver for which the optimality of solutions can be efficiently verified.
* The paper is generally straightfoward to follow and the proofs do not seem to have any obvious mistakes.
* The authors motivate the proposed SDP relaxation well.

**Weaknesses:**

* The empirical evaluation is limited in scope and done entirely with synthetic data. In particular, the authors make the claim that the SDP relaxation frequently computes globally optimal solutions, but I feel the experiments are not extensive enough to fully support such a claim. A more comprehensive experimental evaluation would be required to better understand the practical applicability and limitations of the proposed method.
* The first experiment (Gaussian matching) is not that compelling to me. In Section 4.1, the experiment only considers up to 30 samples in each distribution and Figure 2a seems to suggest the possibility that Frank-Wolfe could perform similarly to the SDP relaxation for larger number of supports while being substantially faster. This is in line with what is reported in Table 2 with the heuristic algorithm, where the gap in performance between SDP and FW diminishes for larger number of supports.
* It seems as though the estimation gap for the first point in Figure 1b is less than 1, though this should not be possible.
* Minor suggestions: 1) The readability of the paper could be enhanced by stating where proofs can be found for each claim, where the claim is presented. For instance, Proposition 3.1 is stated without mentioning where the proof can be found, and similarly for the theorems in appendix B. 2) I assume the numbers in parenthesis in Table 1 are standard errors based on Table 2. It would be helpful to state this in the caption.

**Questions:**

* The authors state at the end of Section 2 that optimal solutions can be found frequently when $m=n$ as shown in Section 4. What about when $m\neq n$? Do the authors find that optimality cannot be verified as frequently in this setting?
* Can the authors provide more details on the choice of parameters in Section 5? How sensitive are the results to these parameter choices?
* Based on the observations I point out above, I am curious if the authors have considered how the behavior of the SDP-based solvers evolves w.r.t. the number of samples and how it compares to existing solvers? What about if the dimensions of the Gaussians is changed?
* The authors extend their SDP relaxation to the task of computing barycenters in appendix D. Have the authors considered comparing (at least qualitatively) their proposed barycenter algorithm to that given in [1] and [2]?
* Would it be possible to explore similar SDP relaxations for variants of the GW problem, such as partial GW [2], outlier-robust GW [3], linear GW [4], etc.? If so, it may be worthwhile to mention in Section 7.


[1] Gromov-wasserstein averaging of kernel and distance matrices, Peyré et al., 2016.
[2] Partial Gromov-Wasserstein Metric, Bai et al., 2024.
[3] Outlier-Robust Gromov-Wasserstein for Graph Data, Kong et al., 2023.
[4] On a linear Gromov-Wasserstein distance, Beier et al., 2022.

**Limitations:**

The main limitation of the proposed work, i.e., high dimensionality of the matrix $mn\times mn$ in SDP, is provided by the authors.

---

> ### Author Rebuttal · Authors · 2024-08-07
>
> *Real data*
>
> We use a publicly available database of triangular meshes (Sumner et al. 2004). We obtain 18 points and compute distance matrices using Dijkstra's algorithm. Each object's probability measure is chosen to be uniform. We apply (GW-SDP) to the corresponding metric-measure spaces to determine the correspondence between the selected vertices across different objects. Two representative examples are given in Figure 1 in the attached pdf file.
>
> Following are the results when we do matching of distances matrices of different objects. In general, we expect shapes of the same animals to have a smaller GW distance than shapes of different animals, which is indeed the case for the three GW formulations. We still notice that GW-SDP consistently returns the smallest value when performing the same matching task.
>
> |               	| GW-SDP   | GW-CG	| eGW-PPA  |
> |-------------------|----------|----------|----------|
> | Elephant-Elephant | 0.007416 | 0.043879 | 0.025688 |
> | Elephant-Cat  	| 0.015695 | 0.050594 | 0.042214 |
> | Cat-Cat       	| 0.006549 | 0.016634 | 0.006757 |
> | Cat-Horse     	| 0.011040 | 0.033736 | 0.011041 |
> | Horse-Horse   	| 0.006287 | 0.033768 | 0.007395 |
>
> *Frank Wolfe with more samples*
>
> This is our interpretation of the Reviewer's comment. Fix both sets of distributions. Increase the number of samples for Frank-Wolfe but fix the number of samples for GW-SDP. What happens?
>
> We conducted this experiment and we noticed that the objective value for Frank-Wolfe (GW-CG in table) decreases as we increase the number of samples and approach the objective value obtained by the SDP relaxation, which is obtained using few samples.
>
> n	GW-SDP	GW-SDP Runtime (s)	GW-CG	GW-CG runtime (s)
> 10	0.4577	6.3753	1.135940	0.000389
> 100			0.629425	0.007571
> 1000			0.540984	2.520011
> 10000			0.496796	138.358954
>
> We noticed that the objective value from Frank-Wolfe is greater than the objective value from GW-SDP. We do not know if this is because the solution from Frank Wolfe is sub-optimal, or if GW-SDP returns a smaller objective than the GW distance between the continuous distributions because of discretization error (from finite samples). This is an interesting question to investigate.
>
> In any case, the experiment suggests it may be possible to mitigate the (lack of) global optimality issues we raise about Frank Wolfe simply by increasing the number of sample points. This is indeed a cheap way. Nevertheless, without the relaxation we propose, it is not possible to tell if the improved objective value obtained by increasing the number of samples is indeed globally optimal. Also, this experiment suggests that the SDP relaxation is able to obtain a far superior objective value with fewer data points compared to Frank Wolfe. One can further investigate trade-offs between these methods (as future work).
>
> We caution against generalizing this approach of increasing samples to attain a better minima too broadly. For Frank Wolfe to converge to the global optimal solution, we need the optimization landscape to be favorable. This might not happen for more complicated data distributions or perhaps graphs. Again, without a relaxation (or a suitable dual problem), one cannot be certain if Frank Wolfe obtains the global solution.
>
> *Approximation gap*
>
> We checked this. The value is numerically equal to one. The plot appears to be less than one due to an issue with the graph plotting software.
>
> *Readability*
>
> Thanks for the suggestion. We will make these changes in the revision.
>
> *$m \neq n$*
>
> We performed an additional experiment where the number of samples in one distribution is fixed ($n=8$) and we vary the number of samples $m$ in the other distribution; see Figure 2 of the attached PDF. We notice that the relaxation is exact whenever $m$ is a multiple of $n$. On the other hand, the relaxation fails to be exact if $m$ is not a multiple of $n$.
>
> *Heuristic solver*
>
> Note that the heuristic solver will not be in the main body of the revised manuscript, based on a suggestion by another Reviewer.
>
> The parameters are chosen heuristically. The parameters $\gamma_2​$ and $\gamma_3$ control the step size in each iteration, while $\gamma_1$​ is used for backtracking to prevent $\gamma_2$​ and $\gamma_3​$ from becoming too large. Through experimentation, we found that the algorithm performs better for relatively small choices.
>
> *Behavior of SDP relaxation when dimensions of the Gaussians change*
>
> We conducted experiments using Gaussian distributions with varying dimensions and found that the performance of the GW-SDP remained consistent across all tested dimensions. As such, we report results for a single representative dimension.
>
> *Barycenter algorithm*
>
> Could the Reviewer clarify which papers [1] and [2] refer to? Thanks.
>
> *Extensions*
>
> It is generally possible to extend the semidefinite relaxation to variants of the GW problem. We briefly describe the semidefinite relaxation to the outlier-robust GW problem by Kong et. al [KLTS:23]. Here, $(X,d_{X})$ and $(Y,d_{Y})$ are two metric spaces with accompanying measures $\mu$ and $\nu$. The distance between $\mu$ and $\nu$ is
>
> $\min~~\langle L, P\rangle+\tau_1 d_{KL} (\pi 1,\alpha) + \tau_2 d_{KL} (\pi^T 1,\beta)$
>
> $\mathrm{s.t.}\left( \begin{array}{cc} P & \mathrm{vec}(\pi)^T \\\ \mathrm{vec}(\pi) & 1 \end{array} \right) \succeq 0 $
>
> $\qquad\sum_{i}P_{(i,j),(k,l)} = f_{j}^{k,l}$, $\Sigma_{j} P_{(i,j),(k,l)} = g_i^{k,l}$
>
> $\qquad P\geq 0$
>
> $\qquad d_{KL}(\mu,\alpha)\leq\rho_1,d_{KL}(\nu,\beta)\leq\rho_2$
>
> Note: The resulting formulation is convex but not an SDP because of the KL divergence. We discuss further extensions in the revised manuscript.
>
> *References*
>
> [KLTS:23] L Kong, J Li, J Tang, & A M-C So. (2023). Outlier-robust Gromov-Wasserstein for Graph Data. In Proceedings of the 37th International Conference on Neural Information Processing Systems.
>
> [SP:04] RW Sumner and J Popovíc. (2004). Deformation transfer for triangle meshes. ACM Transactions on Graphics.

---

> > ### Comment · Reviewer_3Vyi · 2024-08-10
> > **Response to rebuttal**
> >
> > I thank the authors for their extensive response, which clarified some of the raised points.
> >
> > Overall, I think the contributions of the paper outweigh its deficiency. The new experiments conducted during the rebuttal period would significantly strengthen the paper. Given these points, I increased my score to Weak Accept.

---

> > > ### Author Response · Authors · 2024-08-12
> > >
> > > Thank you for going through the rebuttal and your vote of confidence.

---

### Official Review · Reviewer_nsV6 · 2024-07-12

**Soundness:** 4
**Presentation:** 3
**Contribution:** 3
**Rating:** 6
**Confidence:** 4

**Summary:**

The authors provide SDP relaxations of the Gromov-Wasserstein distance, which turns out to provide global optimal in many cases.

**Strengths:**

-- The SDP relaxations and proofs of their exactness are most elegant.

-- The authors suggest that their method does not make as strong assumptions on the loss as in the case of the previous work (e.g., the Proximal Point algorithm).

**Weaknesses:**

Commonly used pythonOT library (https://pythonot.github.io/auto_examples/gromov/plot_gromov.html) implements the conditional Gradient algorithm (ot.gromov.gromov_wasserstein), the Proximal Point algorithm with Kullback-Leibler as proximal operator (ot.gromov.entropic_gromov_wasserstein), and the  Projected Gradient algorithm with entropic regularization (ot.gromov.entropic_gromov_wasserstein), but the authors do not compare their run-time against neither of the methods. Plausibly, this is because the run-time of the SDP solver is much higher?

**Questions:**

How does the runtime compare with that of commonly used algorithms?

If the SDP relaxations are correct, it should be possible to derive first-order methods for these, which could perhaps be simplified substantially, compared to general-purpose SDP solvers. Have you explore the direction?

**Limitations:**

The run-time is not discussed.

---

> ### Author Rebuttal · Authors · 2024-08-07
>
> *Run time comparisons*
>
> We wish to clarify that the current manuscript compares run times of the Conditional Gradient (CG) method, the entropic Gromov-Wasserstein method and our method. The comparisons are found in Page 5 of the submitted manuscript. For a fixed number of samples, the SDP runtime is far more costly than the other two algorithms.
>
> We performed additional experiments to compare the runtime with algorithms suggested by the Reviewer. These results will be included in the revised manuscript.
>
> | m  | GW-SDP 	| GW-CG	| eGW-PGD  | eGW-PPA  |
> |----|------------|----------|----------|----------|
> | 8  | 0.557457   | 0.000349 | 0.162370 | 2.625325 |
> | 12 | 18.970195  | 0.000460 | 0.042703 | 2.116204 |
> | 16 | 17.000997  | 0.000426 | 0.048222 | 3.316097 |
> | 20 | 108.477784 | 0.000556 | 0.056619 | 2.986389 |
> | 24 | 56.620260  | 0.000523 | 0.067252 | 4.503040 |
>
> We also conducted additional experiments where we compared the performance of GW-SDP with GW-CG and eGW-PPA. (These are described in the global comment and in the attached PDF.) Interestingly, in the experiment with real data, the objective values obtained from eGW-PPA are smaller than the values obtained from GW-CG, though still larger than those obtained by GW-SDP. (A smaller objective value is better in this instance.) This suggests the GW-SDP is the best algorithm for finding optimal solutions (in terms of the smallest objective value), followed by the eGW-PPA and then GW-CG.
>
> *First-order methods*
>
> First-order methods should improve the run-time in general. We investigated this approach but encountered numerous difficulties.
>
> It is difficult to apply most first-order methods for solving SDPs to our relaxation. The main reason is that our formulation contains many constraints whereas most numerical techniques typically rely on having few affine constraints.
>
> There is a line of work that uses proximal methods in combination with numerical schemes that allows families of constraints to be decoupled such as the ADMM method. Unfortunately, these ideas also do not translate easily. Our formulation has many different types of constraints: PSD, doubly stochasticity, non-negativity, and marginal sums. It is not clear how one designs proximal operators that project onto the intersection of these constraints, or design ADMM schemes that combine several different proximal operators.
>
> There are some prior works that develop numerical schemes for solving semidefinite relaxations for the closely related Quadratic Assignment Problem (QAP) [KKBL:15,DML:17,OWX:18]. These works do point out that these semidefinite relaxations, while powerful, are not easy to solve. This suggests that it is equally difficult to develop numerical schemes for our semidefinite relaxations.
>
> In fact, the scale of the problems we work on are not far from state-of-the-art for semidefinite relaxations of the QAP. In a work by Oliveira et al. -- a work purely about numerical optimization -- the authors propose an ADMM scheme to solve QAP instances where the dimension is 40 while the largest GW instance we solve has dimension m=n=32. This means that our problem size is not far from the state of the art.
>
> We hope that the Reviewer accepts and values the conceptual and theoretical contributions already present in this paper, and recognizes that the scope of developing scalable numerical schemes, while important, goes comfortably beyond the scope of a single paper.
>
> *References*
>
> [KKBL:15] I. Kezurer, S. Z. Kovalsky, R. Basri, & Y. Lipman. (2015). Tight relaxation of quadratic match. In Computer Graphics Forum, volume 34, pages 115–128. Wiley Online Library.
>
> [DML:17] N. Dym, H. Maron, & Y. Lipman. (2017). DS++: A Flexible, Scalable and Provably Tight Relaxation for Matching Problems. ACM Transactions on Graphics, 36(184):1–14.
>
> [OWX:18] D. E. Oliveira, H. Wolkowicz, & Y. Xu, (2018). ADMM for the SDP relaxation of the QAP. Mathematical Programming Computation, 10(4), 631-658.

---

> > ### Comment · Reviewer_nsV6 · 2024-08-10
> > **Thank you!**
> >
> > Many thanks for the clarifications.
> >
> > I believe that Mathias Staudigl, Shimrit Shtern, and Pavel Dvurechensky are working on related first-order methods suitable for problems with many constraints and possibly non-linear (e.g. entropy) objectives, but I cannot provide the pre-print at the moment.
> >
> > I should like to say I also find the response to reviewer fpsx (with the claims of global optimality unquantified) underwhelming.

---

> > > ### Author Response · Authors · 2024-08-12
> > >
> > > >I believe that Mathias Staudigl, Shimrit Shtern, and Pavel Dvurechensky are working on related first-order methods suitable for problems with many constraints and possibly non-linear (e.g. entropy) objectives, but I cannot provide the pre-print at the moment.
> > >
> > > Thank you for the reference. We will look out for it.
> > >
> > > >I should like to say I also find the response to reviewer fpsx (with the claims of global optimality unquantified) underwhelming.
> > >
> > > When writing the rebuttal, we intended to write a quick summary before moving on to the other points in the rebuttal. The claims of global optimality were to be understood under the conditions that the approximation ratio is one. This was a point we made in the manuscript. When writing the rebuttal, it did not occur to us that it was necessary to repeat these conditions. Unfortunately, the rebuttal may have come across as overstating an incorrect claim. That is not our intention, and we apologize for the confusion.
> > >
> > > To be clear, the claims of global optimality occurs only when the approximation ratio is one.
> > >
> > > (In the follow-up replies to Reviewer fpsx you will find the follow-up exchange on this topic.)

---

### Author Rebuttal · Authors · 2024-08-07

We thank the Reviewers for taking time to provide valuable feedback. We have incorporated many of these suggestions with additional experiments which we believe improve the paper substantially. We outline some of these new experiments and contributions below, and describe them in further detail to the corresponding Reviewer who made these suggestions.

We are grateful that the Reviewers appreciate the conceptual contributions of our work, namely, that it computes global optimal solutions of the GW problem with a certificate of optimality. All Reviewers raised concerns about the run-time -- this is understandable. We too share the same concerns and are actively working on this aspect. We describe some of the technical difficulties below. Nevertheless, our work is the only work we are aware of that addresses global optimality, and we hope the Reviewers consider such a contribution valuable in spite of the run-time concerns.

We provide below general responses for common concerns questions. For the more detailed answer to each reviewer's question, we provide the answer as a comment following each of the individual reviews.

*Experiment on real data and more baseline solvers*

We use a publicly available database of triangular meshes (Sumner et al. 2004). We obtain 18 points and compute distance matrices using Dijkstra's algorithm. Each object's probability measure is chosen to be uniform. We apply (GW-SDP) to the corresponding metric-measure spaces to determine the correspondence between the selected vertices across different objects. Two representative examples are given in Figure 1 in the attached pdf file.

Following are the results when we do matching of distances matrices of different objects. In general, we expect shapes of the same animals to have a smaller GW distance than shapes of different animals, which is indeed the case for the three GW formulations. We still notice that GW-SDP consistently returns the smallest value when performing the same matching task.

|               	| GW-SDP   | GW-CG	| eGW-PPA  |
|-------------------|----------|----------|----------|
| Elephant-Elephant | 0.007416 | 0.043879 | 0.025688 |
| Elephant-Cat  	| 0.015695 | 0.050594 | 0.042214 |
| Cat-Cat       	| 0.006549 | 0.016634 | 0.006757 |
| Cat-Horse     	| 0.011040 | 0.033736 | 0.011041 |
| Horse-Horse   	| 0.006287 | 0.033768 | 0.007395 |

*Experiment where $m \neq n$*

We performed an additional experiment where the number of samples in one distribution is fixed ($n=8$) and we vary the number of samples $m$ in the other distribution; see Figure 2 of the attached PDF. We notice that the relaxation is exact whenever $m$ is a multiple of $n$. On the other hand, the relaxation fails to be exact if $m$ is not a multiple of $n$. For the runtime, please check the results in Table 1 of the pdf.

*Applying Frank-Wolfe with more sample points*

Reviewer 3Vyi raised an interesting suggestion to increase the number of samples for GW-CG (non-convex GW solver using conditional gradient descent or Frank-Wolfe algorithm) vs our GW-SDP solver for a fixed number of samples. We noticed that the objective value for GW-CG decreases as we increase the number of samples. For 100000 sample points, the GW-CG algorithm is more expensive and has a poorer objective value than our method with 10 sample points. This suggests that our method can give good approximations of the GW distance with fewer sample points than existing methods.

| n 	| GW-SDP | GW-SDP Runtime (s) | GW-CG	| GW-CG runtime (s) |
|-------|--------|--------------------|----------|-------------------|
| 10	| 0.4577 | 6.3753         	| 1.135940 | 0.000389      	|
| 100   |    	|                	| 0.629425 | 0.007571      	|
| 1000  |    	|                	| 0.540984 | 2.520011      	|
| 10000 |    	|                	| 0.496796 | 138.358954    	|

*Prohibitive run time of the GW-SDP solver and potential improvements*

We acknowledge the concerns about the run time of the GW-SDP.

We are actively trying to develop faster algorithms. Our current manuscript suggests some potential directions. Reviewer nsV6 suggests first order methods and we discuss this in detail in our response to Reviewer nsV6. In short, many existing techniques do not apply. In fact, in the work by Oliveira et al. on semidefinite relaxations on the closely related QAP, the authors solve problem instances of size 40. In our experiments, the largest instance is 32, and hence not far off state of the art.

Nevertheless, we hope the Reviewer appreciates the important conceptual contributions of our work. It is the only work we are aware of that addresses global optimality in the GW problem.

*References*

[OWX:18] D. E. Oliveira, H. Wolkowicz, & Y. Xu, (2018). ADMM for the SDP relaxation of the QAP. Mathematical Programming Computation, 10(4), 631-658.

[SP:04] RW Sumner and J Popovíc. (2004). Deformation transfer for triangle meshes. ACM Transactions on Graphics.

---

### Decision · Program_Chairs · 2024-09-25

**Decision:**

Accept (poster)

**Comment:**

The paper received mixed, but mostly positive reviews. Reviewers note an elegant and new formulation that allows global optimiyation, and elegant proofs, solid and interesting theoretical aspects and good experimental results in terms of objective values of the optimization problem. On the downside limited evaluation on mostly synthetic examples and high runtime are noted. The rebuttal is positively acknowledged.